# EarthquakeNPP: A Benchmark for Earthquake Forecasting with Neural Point Processes

**Samuel Stockman**                                   *sam.stockman@bristol.ac.uk*
*School of Earth Sciences*
*University of Bristol, UK*

**Daniel Lawson**                                     *dan.lawson@bristol.ac.uk*
*School of Mathematics*
*University of Bristol, UK*

**Maximilian Werner**                                 *max.werner@bristol.ac.uk*
*School of Earth Sciences*
*University of Bristol, UK*

**Reviewed on OpenReview:** *https://openreview.net/forum?id=dIcNAg6ZuZ*

## Abstract

For decades, classical point process models, such as the epidemic-type aftershock sequence (ETAS) model, have been widely used for forecasting the event times and locations of earthquakes. Recent advances have led to Neural Point Processes (NPPs), which promise greater flexibility and improvements over such classical models. However, the currently-used benchmark for NPPs does not represent an up-to-date challenge in the seismological community, since it contains data leakage and omits the largest earthquake sequence from the region. Additionally, initial earthquake forecasting benchmarks fail to compare NPPs with state-of-the-art forecasting models commonly used in seismology. To address these gaps, we introduce EarthquakeNPP: a benchmarking platform that curates and standardizes existing public resources: globally available earthquake catalogs, the ETAS model, and evaluation protocols from the seismology community. The datasets cover a range of small to large target regions within California, dating from 1971 to 2021, and include different methodologies for dataset generation. Benchmarking experiments, using both log-likelihood and generative evaluation metrics widely recognised in seismology, show that none of the five NPPs tested outperform ETAS. These findings suggest that current NPP implementations are not yet suitable for practical earthquake forecasting. Nonetheless, EarthquakeNPP provides a platform to foster future collaboration between the seismology and machine learning communities.

## 1 Introduction

Operational earthquake forecasting by global governmental organisations such as the US Geological Survey (USGS) necessitates the development of models which can forecast the times and locations of damaging earthquakes. While model development is ongoing in the seismology community, recent progress has relied upon refinement of a spatio-temporal point process model known as the Epidemic-Type Aftershock Sequence (ETAS) model (Ogata, 1988; 1998). This continued reliance on a low-dimensional parametric framework stands in contrast to the substantial growth in available earthquake data (Takanami et al., 2003; Shelly, 2017; Ross et al., 2019; White et al., 2019; Mousavi et al., 2020; Tan et al., 2021; Mousavi & Beroza, 2023).

In contrast, the machine learning community has offered promising advancements over classical point process models like ETAS with Neural Point Process (NPP) models, showcasing greater flexibility (Du et al., 2016; Omi et al., 2019a; Shchur et al., 2019; Jia & Benson, 2019; Chen et al., 2021; Zhou et al., 2022; Zhou & Yu, 2024). While some initial benchmarking of these models has been conducted on an earthquake dataset in Japan, these experiments lack relevance for stakeholders in the seismology community. The benchmark omits the largest earthquake sequence from the region, introduces data leakage with non-sequential train-test splits, and does not compare against state-of-the-art models like ETAS.

Here, we introduce EarthquakeNPP: a curated collection of datasets designed for benchmarking NPP models in earthquake forecasting, accompanied by a state-of-the-art benchmark model. These datasets are derived from publicly available raw data, which we process and configure within our platform to facilitate meaningful forecasting experiments relevant to stakeholders in the seismology community. Covering various regions of California, these datasets represent typical forecasting zones and encompass data commonly utilized by forecast issuers. Moreover, employing modern techniques, some datasets include smaller magnitude earthquakes, exploring the potential of numerous small events to enhance forecasting performance through flexible NPPs. To unify efforts, we present an operational-level implementation of the ETAS model alongside the datasets, serving as the benchmark for NPPs.

Although initial benchmarking shows that none of the five tested NPP implementations outperform ETAS, EarthquakeNPP is designed to support ongoing model development and evaluation. In addition to the standard log-likelihood metric common in the NPP literature, the platform incorporates the generative evaluation procedures used in seismology for more rigorous benchmarking. This ensures that future NPPs (and other models such as time series approaches (Wang et al., 2017) and Bayesian point processes (Serafini et al., 2023)) can have direct relevance to seismological stakeholders. All datasets, experiments, and documentation are available at `https://github.com/ss15859/EarthquakeNPP`.

## 1.1 Related Work

### 1.1.1 Benchmarking by the NPP Community

Chen et al. (2021) introduced an earthquake dataset for benchmarking the Neural Spatio-temporal Point Process (NSTPP) model using a global dataset from the U.S. Geological Survey, focusing on Japan from 1990 to 2020. They considered earthquakes with magnitudes above 2.5, splitting the data into month-long segments with a 7-day offset. They exclude earthquakes from November 2010 to December 2011, deeming these sequences "too long" and "outliers". However, this period includes the 2011 Tohoku earthquake (Mori et al., 2011), the largest earthquake recorded in Japan and the fourth largest in the world, at magnitude 9.0. This exclusion renders the benchmarking experiment irrelevant for seismologists, as it is precisely these large earthquakes and their aftershocks that are crucial to forecast due to their damaging impact.

The dataset is partitioned into training, testing, and validation segments. Rather than following a chronological split that would reflect operational forecasting, the segments are assigned in an alternating pattern. This design introduces *data leakage*, as it misrepresents a realistic forecasting setup and artificially inflates performance measures due to the nature of earthquake triggering (Freed, 2005). Specifically, because the model is evaluated on windows that immediately precede its training windows, it can exploit backward-in-time causal dependencies. Section B.2 quantifies the resulting performance inflation, expressed in terms of information gain.

Although earthquakes with magnitudes above 2.5 are considered by Chen et al. (2021), following a change in USGS policy on global data collection, from 2009 onwards, only events above magnitude 4.0 are recorded in the dataset. For earthquake forecasting in Japan, seismologists use datasets from Japanese data centers since they are more comprehensive and complete than global datasets. Section A.2 describes the biases incurred from such data missingness.

Chen et al. (2021) benchmark their model against another spatio-temporal model, Neural Jump SDEs (Jia & Benson, 2019), and a temporal-only Hawkes process, even though a spatio-temporal Hawkes process would provide a more rigorous benchmark. Subsequent papers adopting this benchmark (Zhou et al., 2022; Yuan

et al., 2023; Zhou & Yu, 2024) similarly lack comparisons to a spatio-temporal Hawkes process, benchmarking instead against temporal-only or spatial-only baselines or other spatio-temporal NPPs.

### 1.1.2 Benchmarking by the Seismology Community.

Model comparison has been crucial in the development of earthquake forecasting models since their inception (Kagan & Knopoff, 1987; Ogata, 1988). The Collaboratory for the Study of Earthquake Predictability (CSEP) (Michael & Werner, 2018; Schorlemmer et al., 2018; Savran et al., 2022; Iturrieta et al., 2024) (https://cseptesting.org/ ) aims to unify the framework for earthquake model testing and evaluation, hosting retrospective and fully prospective forecasting experiments globally. CSEP benchmarks short-term models using performance metrics that require forecasts to be generated by simulating many repeat sequences over a specified time horizon (typically one day). These simulated forecasts are compared by discretizing time and space intervals, with test statistics calculated for event counts, magnitudes, locations, and times. The simulation-based approach allows the inclusion of generative models that do not output explicit earthquake probabilities (i.e., a likelihood), and enables evaluation of the full distribution of entire sampled sequences.

Two existing works benchmark NPPs for earthquake forecasting within the seismology community. The first by Dascher-Cousineau et al. (2023) extends a temporal-only NPP from Shchur et al. (2019) to include earthquake magnitudes. The second by Stockman et al. (2023) extends another temporal-only model by Omi et al. (2019a) to target larger magnitude events. Both models are benchmarked against a temporal ETAS model, showing moderate improvements over the baseline. Extending these models to include spatial data is necessary for further testing and potential operational use in the seismological community.

## 2 Background

### 2.1 Spatio-Temporal Point Processes

A spatio-temporal point process is a continuous-time stochastic process that models the random number of events $N(S \times (t_a, t_b])$ which occur in a space-time interval $\mathcal{S} \times (t_a, t_b]$, $\mathcal{S} \subseteq \mathbb{R}^2$, $(t_a, t_b] \subset \mathbb{R}^+$. This process is typically defined by a non-negative *conditional intensity function*

$$\lambda(t, \mathbf{x}|\mathcal{H}_t) := \lim_{\Delta t, \Delta \mathbf{x} \to 0} \frac{\mathbb{E}\left[N([t, t + \Delta t) \times B(\mathbf{x}, \Delta \mathbf{x})|\mathcal{H}_t]\right]}{\Delta t |B(\mathbf{x}, \Delta \mathbf{x})|}, \tag{1}$$

where $\mathcal{H}_t = \{(t_i, \mathbf{x}_i)|t_i < t\}$ denotes the history of events preceding time $t$ and $|B(\mathbf{x}, \Delta \mathbf{x})|$ is the Lebesgue measure of the ball $B(\mathbf{x}, \Delta \mathbf{x})$ with radius $\Delta \mathbf{x}$. Given we observe a history of events up to $t_i$, the probability density function (pdf) of observing an event at time $t$ and location $\mathbf{x}$ is given by

$$p(t, \mathbf{x}|\mathcal{H}_{t_i}) = \lambda(t, \mathbf{x}|\mathcal{H}_{t_i}) \cdot \exp\left(-\int_{t_i}^{t} \int_{\mathcal{S}} \lambda(s, \mathbf{z}|\mathcal{H}_s) d\mathbf{z} ds\right). \tag{2}$$

Most models specify the conditional intensity function, though some (e.g. Shchur et al., 2019; Chen et al., 2021; Yuan et al., 2023) directly model this pdf. Model parameters are typically estimated by maximizing the log-likelihood of observed events within a training time interval $[T_0, T_1]$ and spatial region $\mathcal{S}$,

$$\log p(\mathcal{H}_T) = \underbrace{\sum_{i=0}^{n} \log \lambda(t_i|\mathcal{H}_{t_i}) - \int_{T_0}^{T_1} \int_{\mathcal{S}} \lambda(s, \mathbf{z}|\mathcal{H}_s) d\mathbf{z} ds}_{\text{Temporal log-likelihood}} + \underbrace{\sum_{i=0}^{n} \log f(\mathbf{x}_i|t_i, \mathcal{H}_{t_i})}_{\text{Spatial log-likelihood}}, \tag{3}$$

where the decomposition of the spatio-temporal conditional intensity function, $\lambda(t_i, \mathbf{x}_i|\mathcal{H}_{t_i}) = \lambda(t_i|\mathcal{H}_{t_i}) \cdot f(\mathbf{x}_i|t_i, \mathcal{H}_{t_i})$, allows the log-likelihood to be written as contributions from the temporal and spatial components. In practice, this exact function is often not maximized directly during training: for models specified through the conditional intensity function, an analytical solution to the integral term is generally not possible and is approximated numerically.

For model evaluation and comparison, the log-likelihood of observing events in the test set can be used as a performance metric. This is consistent with a wealth of literature in the seismology community (see Zechar

et al., 2010, and references therein) as well as the wider general point process literature (Daley & Vere-Jones, 2004), which now includes neural point processes (Shchur et al., 2021). The metric evaluates models that output probability distributions over their predictions and consequently penalises models that are overconfident. Although evaluating on events in the test set, the test log-likelihood, $\log p\left((t_i, \mathbf{x}_i)|t_i \in [T_2, T_3], \mathcal{H}_{T_2}\right)$, may still contain dependence upon events prior to the test window $[T_2, T_3]$, typically contained in the history $\mathcal{H}_{T_2}$ of the intensity function. Comparing the difference in mean log-likelihood per event provides the *information gain* from one model to another (Daley & Vere-Jones, 2004).

Point processes are the dominant modeling approach in the seismology community, used extensively in both real-time operational earthquake forecasting (Mizrahi et al., 2024a) and established benchmarking experiments (CSEP) (Taroni et al., 2018; Rhoades et al., 2018). The point process representation of earthquake data aligns naturally with their occurrence as discrete events in time (Kagan, 1994). Furthermore, this modeling approach is favored over discretized forecasting models (e.g., time series) because it eliminates the need for optimizing binning strategies and allows for immediate updates, rather than waiting until the end of a time bin - a delay that could miss critical, potentially damaging events.

## 2.2 ETAS

The Epidemic Type Aftershock Sequence (ETAS) model (Ogata, 1998) is a spatio-temporal Hawkes process Hawkes (1971); Siviero et al. (2024); Bernabeu & Mateu (2025) which models how earthquakes cluster in time and space. It has been adopted for operational earthquake forecasting by government agencies in California (Milner et al., 2020), New-Zealand (Christophersen et al., 2017), Italy (Spassiani et al., 2023), Japan (Omi et al., 2019b) and Switzerland (Mizrahi et al., 2024b), and performs consistently well in CSEP's retrospective and fully prospective forecasting experiments (e.g. Woessner et al., 2011; Rhoades et al., 2018; Taroni et al., 2018; Cattania et al., 2018; Mancini et al., 2019; 2020; 2022). The general formulation of the model is

$$\lambda(t, \mathbf{x}|\mathcal{H}_t; \theta) = \mu + \sum_{i:t_i < t} g(t - t_i, ||\mathbf{x} - \mathbf{x}_i||_2^2, m_i), \tag{4}$$

where $\mu$ is a constant background rate of events, $g(\cdot, \cdot, \cdot)$ is a non-negative excitation kernel which describes how past events contribute to the likelihood of future events and $m_i$ are the associated magnitudes of each event. The equivalent formulation as a Hawkes branching process accompanies a causal branching structure $\mathbf{B}$. This concept broadly aligns with the understanding of the physics of earthquake triggering and interaction, e.g. via dynamic wave triggering (Brodsky & van der Elst, 2014) and static stress triggering (Gomberg, 2018; Mancini et al., 2020).

Although ETAS can be fit by maximizing the log-likelihood function directly, parameter estimation is typically performed by simultaneously estimating the branching structure $\mathbf{B}$. Veen & Schoenberg (2008) developed an Expectation Maximisation (EM) procedure, which maximises the marginal likelihood over the unobserved branching structure, $\log \int p(\mathcal{H}_{T_1}|\mathbf{B}, \theta)p(\mathbf{B}|\theta)d\mathbf{B}$ through the iteration

$$\theta^{(k+1)} = \arg\max_{\theta} \mathbb{E}_{\mathbf{B} \sim p(\cdot|\mathcal{H}_{T_1}, \theta^{(k)})}\left[\log p(\mathcal{H}_{T_1}, \mathbf{B}|\theta)\right]. \tag{5}$$

This avoids the need to numerically approximate the integral term in the likelihood, provides more stability during estimation, and simultaneously distinguishes background events from triggered events.

The formulation of the ETAS model we present in the EarthquakeNPP benchmark is implemented in the `etas` python package by Mizrahi et al. (2022). It defines the triggering kernel as

$$g(t, r^2, m) = \frac{e^{-t/\tau} \cdot k \cdot e^{a(m-M_c)}}{(t+c)^{1+\omega} \cdot \left(r^2 + d \cdot e^{\gamma(m-M_c)}\right)^{1+\rho}}, \tag{6}$$

where $r^2$ is the squared distance between events and $k, a, c, \omega, \tau, d, \gamma, \rho$ are the learnable parameters along with the constant background rate $\mu$. This triggering kernel is derived from statistical distributions found through decades of observational studies (Utsu & Seki, 1955; Utsu, 1970; Utsu et al., 1995) and several of the learnable parameters have been linked to physical properties of the earthquake rupture process (Utsu et al., 1995; Ide, 2013).

Despite its widespread use, it is commonly accepted that ETAS is a misspecified model of seismicity. By construction, ETAS describes only self-exciting triggering behaviour and therefore cannot capture inhibitory effects or stress relaxation processes, such as those represented by stress-release models (Zheng & Vere-Jones, 1991; Xiaogu & Vere-Jones, 1994; Bebbington & Harte, 2003), or by models based on elastostatic stress transfer and Coulomb rate-and-state friction (Dieterich, 1994). In addition, foreshock activity has been observed to deviate from ETAS assumptions, both spatially and temporally (McGuire et al., 2005; Brodsky, 2011; Lippiello et al., 2012; Ogata & Katsura, 2014). Finally, to simplify inference, ETAS typically assumes isotropic spatial triggering kernels, despite observational evidence for anisotropic and fault-aligned aftershock distributions (Page & van der Elst, 2022). Together, these limitations motivate the exploration of more flexible modelling frameworks capable of capturing richer spatio-temporal structure in earthquake sequences.

### 2.3 Neural Point Processes

Neural point processes (NPPs) have emerged in recent years within the machine learning literature as flexible alternatives to classical parametric point process models. Their central motivation is to replace restrictive, hand-crafted parametric forms with neural network based components that can learn complex, non-linear dependencies directly from data. This makes them particularly appealing for earthquake forecasting to overcome the known limitations of the ETAS model discussed in Section 2.2.

Early developments focused on temporal point processes (Shchur et al., 2021). Du et al. (2016) introduced the use of recurrent neural networks (RNNs) to encode the event history into a fixed-dimensional latent state, replacing explicit summation over past events with a learned representation of temporal dependence. Subsequent work explored alternative sequence encoders, including LSTMs (Mei & Eisner, 2017) and Transformers (Zuo et al., 2020), alongside a variety of decoding strategies for modelling the conditional intensity or inter-event time distribution (Du et al., 2016; Omi et al., 2019a; Shchur et al., 2019). In most cases, model parameters are learned by maximising the log-likelihood of observed event sequences, although alternative training objectives have also been proposed (Xiao et al., 2017; Li et al., 2018).

These temporal formulations were later extended to spatio-temporal settings (Mukherjee et al., 2025) by incorporating event locations into the history encoder and introducing flexible decoders for the conditional spatial distribution of future events. Existing spatio-temporal NPPs can be broadly grouped into three modelling classes. The first class extends Hawkes-type formulations by replacing parametric triggering kernels with neural network based influence functions, allowing non-stationary and history-dependent excitation (Zhou et al., 2022; Dong et al., 2022; Zhou & Yu, 2024). A second class models event dynamics in continuous time using neural ordinary differential equations, jointly evolving latent temporal states and spatial distributions (Jia & Benson, 2019; Chen et al., 2020). A third, more recent class adopts fully generative approaches based on diffusion or score matching, learning the joint spatio-temporal distribution of events without explicitly parameterising an intensity function (Yuan et al., 2023; Li et al., 2023; Lüdke et al., 2024). These approaches differ substantially in their computational cost, interpretability, and suitability for simulation and likelihood-based evaluation; see Mukherjee et al. (2025) for a detailed discussion.

## 3 EarthquakeNPP Datasets

The EarthquakeNPP datasets (Figure 1) encompass earthquake records, including timestamps, geographical coordinates, and magnitudes, documented within California from 1971 to 2021. California, with its dense network and high seismic hazard, has been extensively studied, demonstrating the utility of forecasting algorithms (Gerstenberger et al., 2004; Field, 2007; Field et al., 2021). It encompasses the San Andreas fault plate boundary system (Zoback et al., 1987) and includes modern high-resolution catalogs with numerous small magnitude earthquakes, offering potential for new, more expressive models.

A central challenge when working with earthquake catalogs is data missingness, referred to in seismology as catalog incompleteness (Mignan & Woessner, 2012). Earthquakes are assumed to be only fully detected above a time- and region-dependent completeness magnitude $M_c$, which reflects limitations of the seismic network and changes in detection capability over time. Ignoring this incompleteness can introduce substantial bias

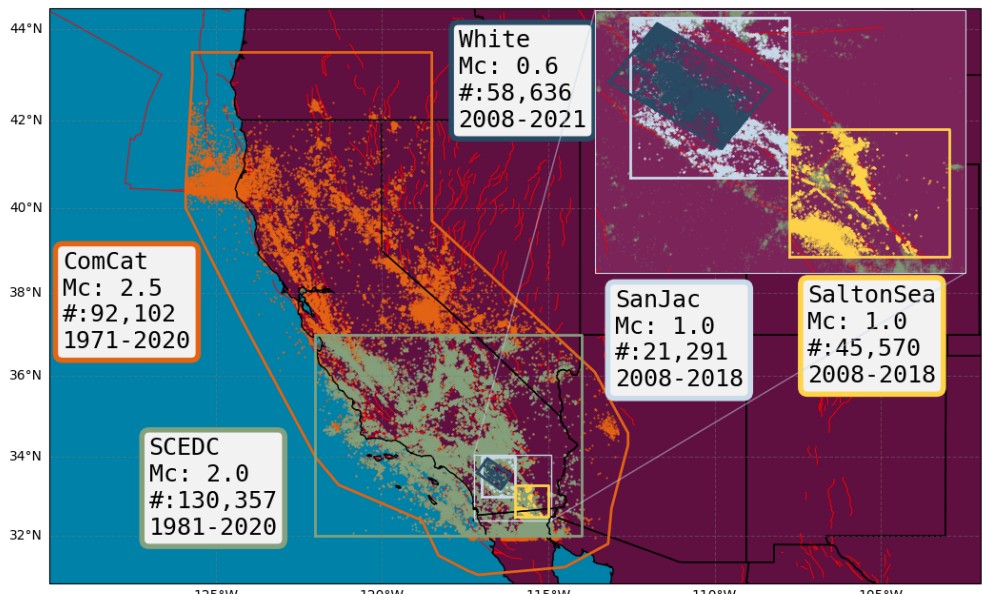

Figure 1: Earthquakes contained in the observational datasets found in EarthquakeNPP. Colours indicate the respective datasets, including the target region, magnitude of completeness $M_c$, number of events and the time period that the dataset spans. In red is a fault map from the GEM Global Active Faults Database (Styron & Pagani, 2020).

into both model fitting and evaluation (Sornette & Werner, 2005), particularly for methods that rely on smaller magnitude events.

All EarthquakeNPP datasets are constructed from publicly available raw catalogs provided by their respective data centres. To enable a consistent and realistic retrospective forecasting experiment, the raw data is preprocessed by restricting each dataset to a target spatial region, truncating events above a dataset-specific magnitude threshold $M_{\text{cut}} \geq M_c$ (e.g., Mignan et al., 2011; Mignan & Woessner, 2012) and removing duplicate locations.

Notebooks to access and preprocess the datasets, along with the associated benchmarking experiments, are publicly available at `https://github.com/ss15859/EarthquakeNPP`, accompanied by detailed dataset documentation. A more in-depth discussion of earthquake catalog generation, completeness, and preprocessing choices is provided in Appendix A. Table 1 summarises the key characteristics of each EarthquakeNPP dataset.

## 4 Benchmarking Experiment

We use EarthquakeNPP to benchmark five spatio-temporal Neural Point Processes (NPPs) against the ETAS model described in Section 2.2. Each of these NPPs has prior positive claims in earthquake forecasting, yet lacks performance comparison with the ETAS model.

**Neural Spatio-Temporal Point Process (NSTPP)** (Chen et al., 2021)**:** A probability density function (pdf)-based NPP that parametrizes the spatial pdf with continuous-time normalizing flows (CNFs). We evaluate their Attentive CNF model due to its superior computational efficiency and overall performance compared to the Jump CNF model (Chen et al., 2021).

**Deep Spatio-Temporal Point Process (DeepSTPP)** (Zhou et al., 2022)**:** A conditional intensity function-based NPP that constructs a non-parametric space-time intensity function driven by a deep latent process. This model features a closed-form intensity function, eliminating the need for numerical approximations.

Table 1: Summary of EarthquakeNPP datasets, including: region, dataset development, magnitude threshold ($\mathbf{M_c}$), number of training (combined with validation) events, and number of testing events. The chronological partitioning of training, validation, and testing periods is also detailed. An auxiliary (burn-in) period begins from the **Start** date, followed by the respective starts of the training, validation, and testing periods. All dates are given as 00:00 UTC on January 1st, unless noted (* refers to 00:00 UTC on January 17th). Finally, we give our purpose for including each dataset.

| | ComCat | SCEDC | White | QTM |
|---|---|---|---|---|
| **Region** | Whole of California | Southern California | San Jacinto Fault-Zone | `QTM_SanJac:` San Jacinto Fault-Zone, `QTM_SaltonSea:` Salton Sea |
| **Development** | The U.S. Geological Survey (USGS) National Earthquake Information Center (NEIC) monitors global earthquakes (Mw 4.5 or larger) and provides complete seismic monitoring of the United States for all significant earthquakes (> Mw 3.0 or felt). Its contributing seismic networks have produced the Advanced National Seismic System (ANSS) Comprehensive Catalog of Earthquake Events and Products. | The Southern California Seismic Network (SCSN) has developed and maintained the standard earthquake catalog for Southern California (Hutton et al., 2010) since the Caltech Seismological Laboratory began routine operations in 1932. Significant network improvements since the 1970s and 1980s reduced the catalog completeness from Mw 3.25 to Mw 1.8. | White et al. (2019) created an enhanced catalog focusing on the San Jacinto fault region, using a dense seismic network in Southern California. This denser network, combined with automated phase picking (STA/LTA), ensures a 99% detection rate for earthquakes greater than Mw 0.6 in a specific subregion (White et al., 2019). | Using data collected by the SCSN, Ross et al. (2019) generated a denser catalog by reanalyzing the same waveform data with a template matching procedure that looks for cross-correlations with the wavetrains of previously detected events. |
| **$\mathbf{M_c}$** | Mw 2.5 | `SCEDC_20:` Mw 2.0, `SCEDC_25:` Mw 2.5, `SCEDC_30:` Mw 3.0 | Mw 0.6 | Mw 1.0 |
| **# Train/Test Events** | 79,037 / 23,059 | `SCEDC_20:` 128,265 / 14,351, `SCEDC_25:` 43,221 / 5,466, `SCEDC_30:` 12,426 / 2,065 | 38,556 / 26,914 | `QTM_SanJac:` 18,664 / 4,837, `QTM_SanJac:` 44,042 / 4,393 |
| **Start-Train-Val-Test-End** | 1971-1981-1998-2007-2020* | 1981-1985-2005-2014-2020 | 2008-2009-2014-2017-2021 | 2008-2009-2014-2016-2018 |
| **Purpose** | Example of data currently in use for operational forecasting (USGS utilizes ComCat in aftershock forecasts they release to the public). | **Three magnitude thresholds (Mw 2.0, 2.5, 3.0 for** `SCEDC_20`, `SCEDC_25`, `SCEDC_30`**)** explore the effect of truncation on forecasting model performance. | To explore if newly detected low magnitude earthquakes contain additional predictive information. | To explore if newly detected low magnitude earthquakes contain additional predictive information (with different detection methodology). |

**Automatic Integration for Spatiotemporal Neural Point Processes (AutoSTPP)** (Zhou & Yu, 2024)**:** A conditional intensity function-based NPP that jointly models the 3D space-time integral of the intensity and its derivative (the intensity function) using a dual network approach.

**Spatio-temporal Diffusion Point Process (DSTPP)** (Yuan et al., 2023)**:** A generative NPP that does not have a likelihood function. DSTPP employs diffusion models to capture complex spatio-temporal dynamics.

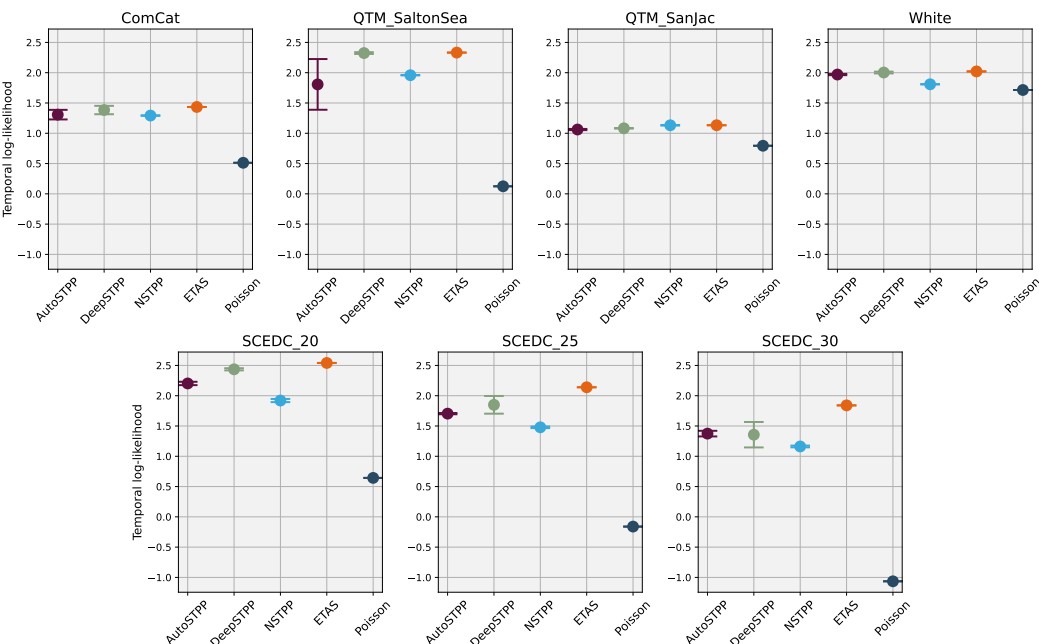

Figure 2: Test temporal log-likelihood scores for all the spatio-temporal point process models on each of the EarthquakeNPP datasets. `SCEDC_20`, `SCEDC_25` and `SCEDC_30` correspond to magnitude thresholds (Mw 2.0, 2.5, 3.0) of the `SCEDC` dataset. Error bars of the mean and standard deviation are constructed for the NPPs using three repeat runs.

**Score Matching-based Pseudolikelihood Estimation of Neural Marked Spatio-Temporal Point Process (SMASH)**(Li et al., 2023)**:** A generative NPP that also lacks a likelihood function. SMASH adopts a normalization-free objective by estimating the pseudolikelihood of marked STPPs through score-matching.

Appendix D provides details on the computational cost of training and inference for all the models tested.

### 4.1 Likelihood Evaluation

Since generating repeated sequences over forecast horizons is computationally costly, the seismology community uses the mean log-likelihood on held-out data for a more streamlined metric during model development (Ogata, 1988; Harte, 2015). Other traditional next-event metrics like Root Mean Square Error (RMSE) and Mean Absolute Error (MAE) are misleading for earthquake forecasting (Hodson, 2022), as earthquake occurrence follows power law distributions (Kagan, 1994; Felzer & Brodsky, 2006) that are heavy-tailed, making the errors non-Gaussian and non-Laplacian, contrary to the assumptions underlying RMSE and MAE (see Section I).

For the three models with valid likelihood functions (NSTPP, DeepSTPP, and AutoSTPP), we present the mean log-likelihood scores in Figures 2 and 3. These scores are compared alongside the ETAS model (Section 2.2) and a homogeneous Poisson process. The Poisson model is fit to events in the auxiliary, training, and validation windows to provide a baseline score for comparison.

Unlike the NPPs, which follow the standard machine learning procedure of training, validation, and testing, ETAS does not typically incorporate validation in its estimation procedure. Thus, it is fit using both the training and validation windows combined. For NPPs, the likelihood for training, validation, and testing depends on events occurring before the respective splits through memory in their history. The exception is NSTPP, which lacks a direct dependency on prior events. Nevertheless, its likelihood is evaluated on the same data as the other models.

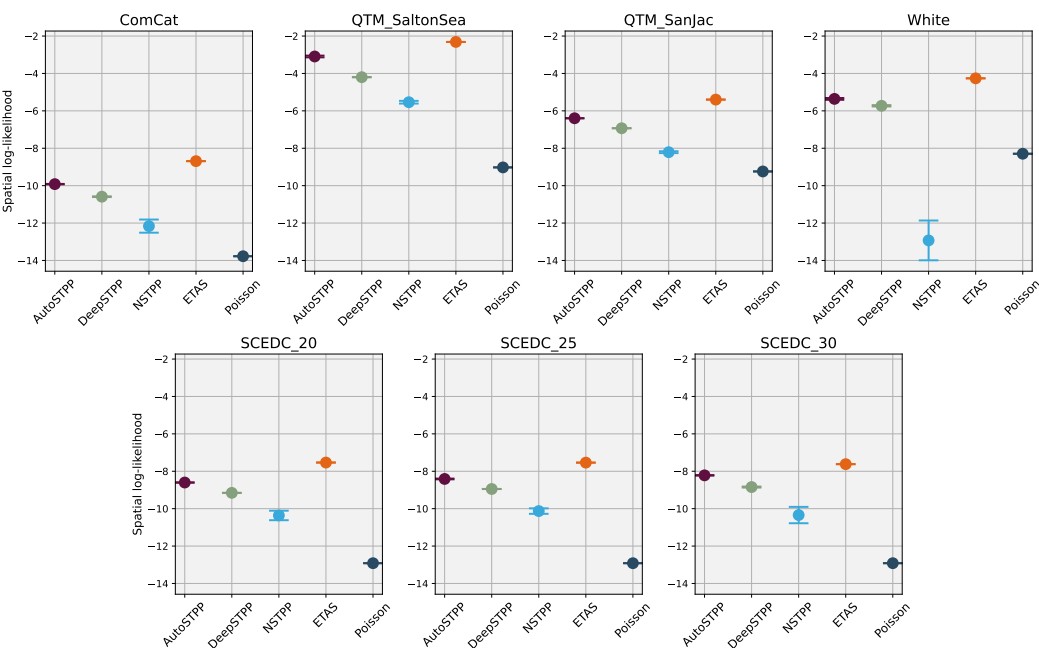

Figure 3: Test spatial log-likelihood scores for all the spatio-temporal point process models on each of the EarthquakeNPP datasets. `SCEDC_20`, `SCEDC_25` and `SCEDC_30` correspond to magnitude thresholds (Mw 2.0, 2.5, 3.0) of the `SCEDC` dataset. Error bars of the mean and standard deviation are constructed for the NPPs using three repeat runs.

To ensure that fitting ETAS on both the training and validation windows does not bias the comparison, we also tested an alternative configuration where ETAS was trained only on the training window. As shown in Appendix C, ETAS performance remains effectively unchanged under this setup. The ETAS formulation (Equation 4) also specifies how the magnitudes of prior earthquakes contribute to the conditional intensity; this magnitude dependence is not implemented in any of the NPPs we benchmark, since it requires modelling choices beyond the scope of this work.

The ETAS model consistently achieves the highest temporal log-likelihood, with NPPs performing comparably or, in some cases, marginally better, except at the higher magnitude thresholds of the `SCEDC` catalog. Among the NPPs, AutoSTPP and NSTPP perform well across several datasets, though their performance is more variable than that of DeepSTPP, which demonstrates consistent, comparable performance to ETAS. Differences in Poisson performance across Figures 2 and 3 are driven by variations in clustering strength, with weakly clustered catalogs appearing nearly Poisson-like and strongly clustered catalogs exhibiting larger departures.

Closer analysis of model performance over time (see Section E) reveals that relative performance to ETAS is poorest during large earthquake sequences. This is likely due to ETAS leveraging the magnitude feature of the data, which enables it to handle these sequences effectively. Conversely, model performance is strongest during "background" periods, when no large earthquakes occur. During these periods, ETAS models the background with a constant rate, while the NPPs improve upon this by capturing the non-stationary nature of the background data. The improved relative temporal performance of all NPPs compared to ETAS, particularly when the magnitude threshold is lowered from 3.0 to 2.0 in the `SCEDC` dataset, indicates that low magnitude earthquakes provide valuable predictive information for NPPs.

ETAS consistently outperforms all NPPs in spatial log-likelihood. Further analysis of model performance over space (see Section E) shows relative performance to ETAS is weakest in the most active and clustered areas (see Figures 12 and 13), likely due to the absence of a magnitude feature in the NPPs. However,

NPPs tend to perform more competitively in regions characterised by spatially complex or diffuse seismicity. AutoSTPP achieves the highest spatial log-likelihood, attributed to its ability to capture anisotropic Hawkes kernels (see Figure 2 of Zhou & Yu (2024)), which are commonly observed in earthquake data (Page & van der Elst, 2022).

## 4.2 CSEP Consistency Tests

EarthquakeNPP supports the earthquake forecast evaluation protocol developed by the Collaboratory for the Study of Earthquake Predictability (CSEP). In this procedure, a model generates 24-hour forecasts through 10,000 repeated simulations of earthquake sequences at the beginning of each day in the testing period. This approach mirrors how earthquake forecasts are produced in operational settings (van der Elst et al., 2022). Models are then evaluated by comparing the observed sequence with the distribution of forecasts generated by the simulations. Four test statistics assess the temporal, spatial, likelihood, and magnitude components of the forecasts. A test is considered failed if the observed statistic falls within a pre-defined rejection region (Figure 14). We apply this evaluation procedure to the two generative NPPs (DSTPP and SMASH) alongside ETAS (Table 2) and present a case study on the 2010 M7.2 El Mayor-Cucapah earthquake, using the forecasts from these models (Figure 4). Appendix F provides an introduction to the CSEP consistency tests, with further details found at `https://cseptesting.org/`, and Appendix G provides further analysis on the simulated forecasts.

Table 2: CSEP consistency tests evaluate the calibration of daily forecasts from three models (ETAS, SMASH, DSTPP) on EarthquakeNPP datasets. A test is performed at the $\alpha = 0.05$ significance level on each day in the testing period. The pass rate indicates the proportion of testing days with non-rejected hypotheses. If the model is the data generator, quantile scores should be uniformly distributed. The KS-Statistic quantifies deviation from uniformity (see Appendix F). ETAS is the only model that forecasts earthquake magnitudes, so is the only model evaluated with the magnitude test.

| Dataset | Model | Number Test | | Spatial Test | | Pseudo Likelihood Test | | Magnitude Test | |
|---|---|---|---|---|---|---|---|---|---|
| | | Pass Rate | KS-Stat. | Pass Rate | KS-Stat. | Pass Rate | KS-Stat. | Pass Rate | KS-Stat. |
| ComCat | ETAS | **95.8%** | 0.222 | **92.0%** | **0.029** | **97.6%** | **0.128** | **93.8%** | **0.113** |
| | SMASH | 86.2% | 0.212 | 68.6% | 0.217 | 87.6% | 0.134 | – | – |
| | DSTPP | 86.7% | **0.116** | 88.6% | 0.070 | 86.3% | 0.138 | – | – |
| SCEDC | ETAS | **92.6%** | **0.347** | **88.3%** | **0.119** | **95.9%** | **0.233** | **90.0%** | **0.256** |
| | SMASH | 69.6% | 0.602 | 51.1% | 0.471 | 68.0% | 0.611 | – | – |
| | DSTPP | 27.4% | 0.840 | 6.1% | 0.935 | 0.8% | 0.988 | – | – |
| QTM_SanJac | ETAS | **95.4%** | 0.151 | **91.7%** | **0.095** | **96.6%** | **0.123** | **94.8%** | **0.076** |
| | SMASH | 81.7% | 0.290 | 55.6% | 0.385 | 53.4% | 0.584 | – | – |
| | DSTPP | 85.7% | **0.110** | 85.7% | 0.240 | 85.3% | 0.136 | – | – |
| QTM_SaltonSea | ETAS | **93.6%** | 0.210 | **90.9%** | 0.206 | **96.4%** | **0.119** | **90.6%** | **0.136** |
| | SMASH | 75.8% | 0.486 | 53.6% | 0.371 | 73.7% | 0.451 | – | – |
| | DSTPP | 87.7% | **0.154** | 88.8% | **0.136** | 85.6% | 0.127 | – | – |
| White | ETAS | **88.3%** | **0.167** | **86.6%** | 0.225 | **90.8%** | **0.233** | **88.8%** | **0.052** |
| | SMASH | 69.3% | 0.274 | 84.5% | **0.150** | 67.7% | 0.246 | – | – |
| | DSTPP | 0.5% | 0.987 | 30.9% | 0.691 | 32.3% | 0.892 | – | – |

ETAS consistently performs best across all datasets and tests. It achieves the highest pass rates and lowest KS statistics, indicating strong calibration and reliability. SMASH shows moderate performance, often outperforming DSTPP but trailing ETAS. Its results vary more across datasets and tests, with occasional strengths (e.g. in White for spatial KS). DSTPP generally performs worse, with lower pass rates and higher KS statistics, especially for the SCEDC and White datasets. However, it achieves relatively good spatial calibration in some cases (e.g., ComCat).

Further analysis of the simulated forecasts in Appendix G provides insight into the consistency test results. Temporally, all models struggle to capture the highest-rate seismicity days, indicating substantial room for improvement in modelling the most hazardous periods. SMASH exhibits highly variable, spiky daily rate forecasts that result in frequent over- and underprediction, while DSTPP produces much smoother

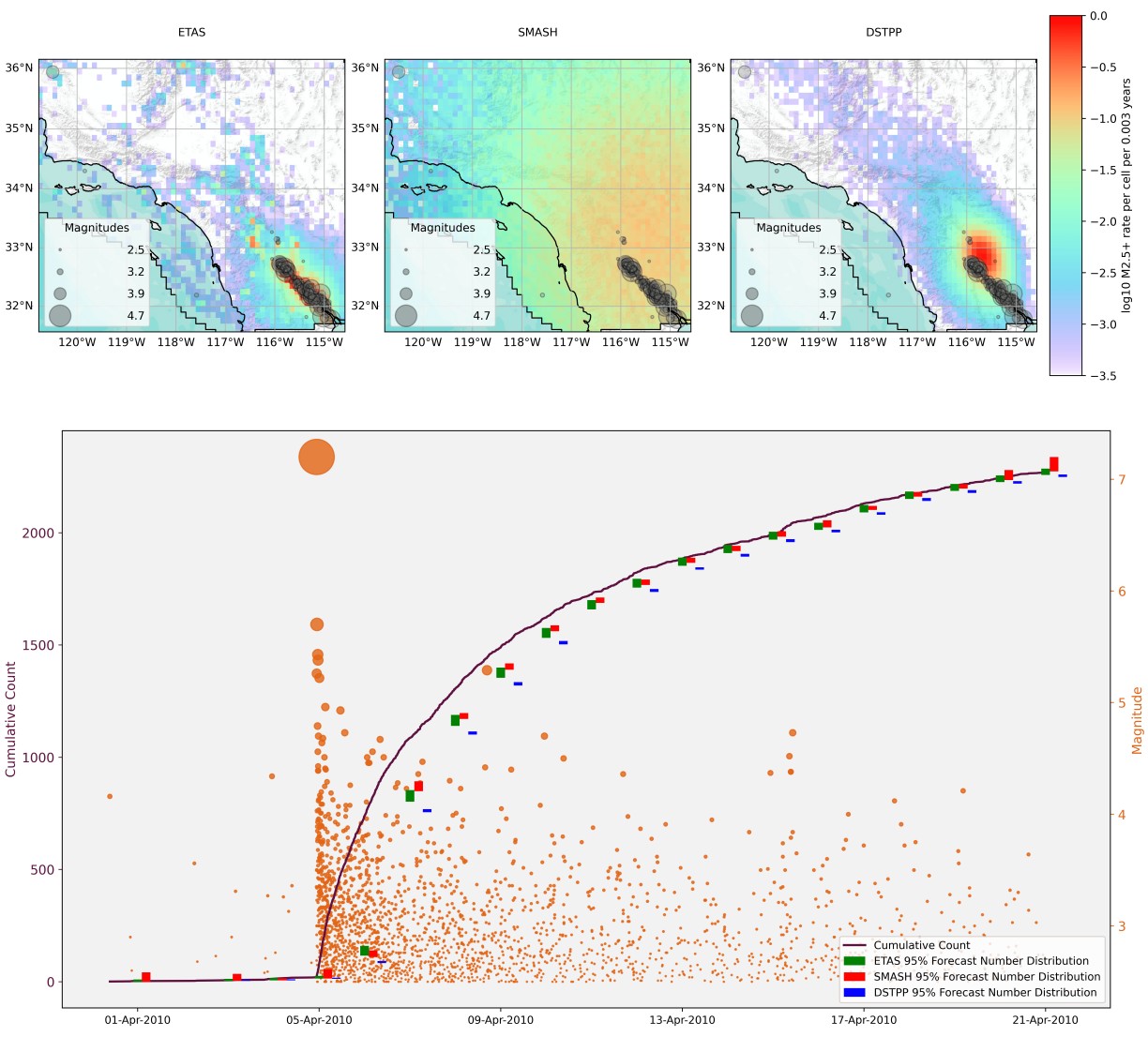

Figure 4: Forecasts from ETAS, SMASH, and DSTPP during the 2010 M7.2 El Mayor-Cucapah earthquake contained in the `ComCat` dataset. Top: Spatial forecasts for the day following the mainshock. ETAS accurately captures the primary aftershock zone along the Laguna Salada fault system. SMASH produces smoother forecasts with broader spatial spread, while DSTPP concentrates its probability mass north of the mainshock epicenter. Bottom: Cumulative earthquake counts over time, with magnitudes shown as scaled orange circles. Forecast number distributions from each model are plotted with 95% confidence intervals. All models initially underestimate aftershock activity. ETAS and SMASH begin to recover after the first week, whereas DSTPP continues to systematically underpredict event counts throughout the sequence.

forecasts that systematically underestimate seismicity across both background and active periods. Spatially, ETAS assigns concentrated rates along known fault structures through its explicit clustering mechanism. In contrast, SMASH generates diffuse spatial forecasts with weak contrast between active and inactive regions, whereas DSTPP more accurately follows fault-aligned structure but often assigns uniformly low spatial rates, particularly in the `SCEDC` and `White` datasets.

We were unable to apply the CSEP evaluation procedure for NSTPP, AutoSTPP and DeepSTPP, since the models are not explicitly formulated to be generative and therefore suffer from slow sampling (see details

in Appendix D). This limitation significantly hinders their ability to be applied to real-time operational earthquake forecasting.

## 5 Discussion and Conclusion

We introduce EarthquakeNPP, a benchmarking platform designed to evaluate Neural Point Process (NPP) models against the state-of-the-art ETAS model for earthquake forecasting. The platform hosts datasets from diverse regions of California, both standard forecasting zones and datasets that incorporate modern detection techniques. We establish two evaluation frameworks tailored to seismology: standard log-likelihood metrics and the generative consistency tests developed by the Collaboratory for the Study of Earthquake Predictability (CSEP), ensuring that successful models can be directly relevant to operational forecasting.

In benchmarking five neural point process (NPP) models against ETAS, we found that none outperformed the baseline, indicating that current NPP architectures are not yet suitable for operational earthquake forecasting. While several NPPs achieve competitive performance during low-activity background periods, they consistently struggle during highly active phases following large earthquakes. Our results highlight several concrete architectural and methodological gaps, which we summarise below as actionable directions for future NPP development.

**Action 1: Encode explicit magnitude dependence.** ETAS explicitly encodes magnitude dependence, whereby larger earthquakes exponentially increase both the rate and spatial extent of subsequent seismicity. None of the benchmarked NPPs incorporate such explicit magnitude scaling, which limits their ability to capture the dominant influence of large events. Future NPP architectures could address this by introducing magnitude-aware design choices, such as hierarchical encodings that distinguish small and large events, magnitude-weighted attention mechanisms, or parameterisations aligned with the logarithmic frequency–magnitude scaling observed in seismicity (Richter, 1935). These approaches would allow NPPs to retain flexibility while incorporating structure that has proven critical for ETAS performance.

**Action 2: Design scalable long-term memory mechanisms.** All evaluated NPPs truncate the conditioning history due to the computational cost of sequence encoders, with models such as DeepSTPP and AutoSTPP conditioning on as few as 20 past events. In contrast, ETAS integrates the full event history, allowing long-past earthquakes, including large or spatially distant events, to influence future rates. Designing NPPs with scalable long-term memory is therefore a critical avenue for improvement. Promising directions include sparse or dilated attention mechanisms (Child et al., 2019; Hassani & Shi, 2022) to reduce quadratic complexity, hierarchical or coarse-to-fine representations of earthquake histories (Yang et al., 2016), and explicit memory compression modules (Kim et al., 2023) that preserve the influence of distant but significant events. Advances in long-context modelling within the NLP literature suggest that such mechanisms are technically feasible (Liu et al., 2025) and may translate naturally to earthquake triggering dynamics.

**Action 3: Align generative training with operational evaluation.** Third, our results reveal a mismatch between how generative NPPs are trained and how they are evaluated. Models such as SMASH and DSTPP are trained to predict or sample the next event, whereas CSEP consistency tests require simulating complete event sequences over fixed forecasting windows. This discrepancy helps explain why some generative models show reasonable short-term accuracy (Yuan et al., 2023; Li et al., 2023) but perform poorly in our multi-event simulation tests. Future generative NPPs may therefore benefit from training objectives that explicitly target long-horizon trajectory behaviour, for example by optimising multi-event simulation losses (e.g. Lüdke et al., 2024) or designing statistics (e.g. equation 13) aligned with the CSEP evaluations used in this study.

**Action 4: Incorporate empirically supported scaling laws.** Finally, our results suggest that the complete removal of physically motivated structure may be counterproductive. While NPPs aim to move beyond parametric models, ETAS kernels encode power-law scaling relationships that are strongly supported by empirical seismology (Kagan, 1994; Felzer & Brodsky, 2006). Hybrid architectures that combine neural density estimation with ETAS-inspired power-law kernels or magnitude-dependent triggering functions may offer a productive middle ground, retaining empirical laws while allowing greater flexibility than purely parametric formulations. For example, replacing Gaussian spatial kernels in existing NPPs (e.g in DeepSTPP) with

learned power-law forms could improve their ability to represent aftershock clustering without sacrificing expressiveness.

EarthquakeNPP, available at `https://github.com/ss15859/EarthquakeNPP`, provides a platform for future NPP developments to be benchmarked against these initial results. The platform is under ongoing development and in the future will see the direct comparison of emerging and other existing models developed within the seismology community, as well as an expansion of datasets included to other seismically active global regions. Successful NPP models on these datasets, for both log-likelihood and CSEP metrics, will be directly impactful to stakeholders in seismology, ultimately enabling their integration into operational earthquake forecasting by government agencies.

### Acknowledgments

This project is funded by Compass - Centre for Doctoral Training in Computational Statistics and Data Science (EPSRC Grant Ref EP/S023569/1). Compass is funded by United Kingdom Research and Innovation (UKRI) through the Engineering and Physical Sciences Research Council (EPSRC), `https://www.ukri.org/councils/epsrc`. This project also has received funding from the European Research Council (ERC) under the European Union's Horizon 2020 research and innovation programme (Grant 821115, Real-time earthquake rIsk reduction for a reSilient Europe (RISE), `http://www.rise-eu.org`) and by United States Geological Survey (USGS) EHP grants G24AP00059 and G25AP00379.

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

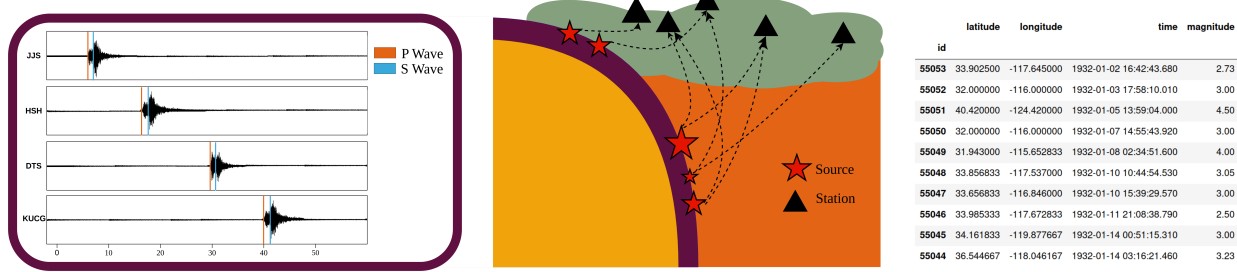

Figure 5: Generating an earthquake catalog involves several key steps: seismic phase picking, magnitude estimation, and the association and location of seismic sources. This process transforms raw waveform data recorded at seismic stations to locations, times, and magnitudes of earthquakes.

Zichao Yang, Diyi Yang, Chris Dyer, Xiaodong He, Alex Smola, and Eduard Hovy. Hierarchical attention networks for document classification. In *Proceedings of the 2016 conference of the North American chapter of the association for computational linguistics: human language technologies*, pp. 1480–1489, 2016.

Yuan Yuan, Jingtao Ding, Chenyang Shao, Depeng Jin, and Yong Li. Spatio-temporal diffusion point processes. In *Proceedings of the 29th ACM SIGKDD Conference on Knowledge Discovery and Data Mining*, pp. 3173–3184, 2023.

J Douglas Zechar, Matthew C Gerstenberger, and David A Rhoades. Likelihood-based tests for evaluating space–rate–magnitude earthquake forecasts. *Bulletin of the Seismological Society of America*, 100(3): 1184–1195, 2010.

Xiao-Gu Zheng and David Vere-Jones. Application of stress release models to historical earthquakes from north china. *Pure and Applied Geophysics*, 135(4):559–576, 1991.

Zihao Zhou and Rose Yu. Automatic integration for spatiotemporal neural point processes. *Advances in Neural Information Processing Systems*, 36, 2024.

Zihao Zhou, Xingyi Yang, Ryan Rossi, Handong Zhao, and Rose Yu. Neural point process for learning spatiotemporal event dynamics. In *Learning for Dynamics and Control Conference*, pp. 777–789. PMLR, 2022.

Weiqiang Zhu and Gregory C Beroza. Phasenet: a deep-neural-network-based seismic arrival-time picking method. *Geophysical Journal International*, 216(1):261–273, 2019.

Mark D Zoback, Mary Lou Zoback, Van S Mount, John Suppe, Jerry P Eaton, John H Healy, David Oppenheimer, Paul Reasenberg, Lucile Jones, C Barry Raleigh, et al. New evidence on the state of stress of the san andreas fault system. *Science*, 238(4830):1105–1111, 1987.

Simiao Zuo, Haoming Jiang, Zichong Li, Tuo Zhao, and Hongyuan Zha. Transformer hawkes process. In *International conference on machine learning*, pp. 11692–11702. PMLR, 2020.

# A Earthquake Catalog Data

## A.1 Earthquake Catalog Generation

Data missingness, referred to in seismology as catalog (in)completeness, is the primary challenge faced with earthquake catalogs. It is an important and unavoidable feature, and is a result of how earthquakes are detected and characterised. Below, we briefly overview the process of generating an earthquake catalog to illustrate the data quality issues. In the subsequent section, we review catalog incompleteness and its potential impact on the performance and evaluation of forecasting models.

**Seismometers and Seismic Networks.** A seismometer is an instrument that detects and records the vibrations caused by seismic waves (Stein & Wysession, 2009; Shearer, 2019). It consists of a sensor to detect ground motion and a recording system to log three-dimensional ground motion over time, typically vertical and horizontal velocities. Seismic networks, comprising multiple seismometers, monitor seismic activity at regional, national or global scales (see, e.g., (Woessner et al., 2010) and references therein). High-density networks with modern, sensitive equipment provide more detailed and accurate data, enhancing the ability to detect and analyse smaller and more distant earthquakes.

**From Waveforms to Phase Picking.** The process of converting raw continuous seismic waveforms into useful earthquake data begins with phase picking, which identifies the arrival times of the primary (P) and secondary (S) waves of an earthquake. Historically, this was done manually, but now automated algorithms, such as the STA/LTA algorithm, detect wave arrivals by analyzing signal amplitude changes (Allen, 1982). Recent algorithms, such as machine learning classifiers (e.g. Zhu & Beroza, 2019; Lapins et al., 2021) and template-matching (e.g. Ross et al., 2019), can process much higher volumes of data efficiently and are often able to detect events of much smaller magnitudes.

**Earthquake Association and Location** After phase picking, the next step is to associate phases from different seismometers with the same earthquake. Simple algorithms require at least four phase arrivals to be detected on different stations within a short time interval to declare an event. Once phases are associated, location estimation determines the earthquake's hypocenter and origin time by minimizing travel-time residuals using linearized or global inversion algorithms (Thurber, 1985; Lomax et al., 2000). Given the potential for misidentified or mis-associated phase arrivals due to low signal-to-noise of small events or the near-simultaneous occurrence during very active aftershock sequences, an automated system typically first picks arrival times and determines a preliminary location, which is subsequently reviewed by a seismologist (e.g. Woessner et al., 2010, and references therein). Locations are typically reported as the geographical coordinates and depths where earthquakes first nucleated (hypocenters), although some catalogs report the centroid location, a central measure of the extended earthquake rupture.

**Earthquake Magnitude Calculation** The magnitude of an earthquake quantifies the energy released at the source and was originally defined in the seminal paper by Richter (1935). The original definition, now referred to as the local magnitude (ML), is calculated from the logarithm of the amplitude of waves recorded by seismometers. This scale, however, "saturates" at higher magnitudes, meaning it underestimates magnitudes for various reasons. This led to introduction of the moment magnitude scale (Mw) (Hanks & Kanamori, 1979), which computes the magnitude based on the estimated seismic moment $M_0$, which can be related to the physical rupture process via

$$M_0 = \text{rigidity} \times \text{rupture area} \times \text{slip}, \tag{7}$$

where rigidity is a mechanical property of the rock along the fault, rupture area is the area of the fault that slipped, and slip is the distance the fault moved. Mw is determined seismologically via a spectral fitting process to the earthquake waveforms. In practice, it can be challenging to use a single magnitude scale for a broad range of magnitudes, therefore a range of scales may be present within a single catalog, and approximate magnitude conversion equations may be used to homogenize the scales (e.g. Herrmann & Marzocchi, 2021, and references therein).

## A.2 Earthquake Catalog Completeness

All of the EarthquakeNPP datasets are made publicly available by their respective data centers in raw format. However, constructing a suitable retrospective forecasting experiment from this raw data requires appropriate pre-processing. This typically involves truncating the dataset above a magnitude threshold $M_{\text{cut}}$ and within a target spatial region to address incomplete data, known as catalog completeness $M_c$ (e.g., Mignan et al., 2011; Mignan & Woessner, 2012).

There are several reasons why an earthquake may not be detected by a seismic network. Small events may be indistinguishable from noise at a single station, or insufficiently corroborated across multiple stations. Another significant cause of missing events occurs during the aftershock sequence of large earthquakes, when the seismicty rate is high (Kagan & Knopoff, 1987; Hainzl, 2022). Human or algorithmic detection abilities

Table 3: Summary of additional datasets, including: magnitude threshold ($\mathbf{M_c}$), number of training events, and number of testing events. The chronological partitioning of training, validation, and testing periods is also detailed. An auxiliary (burn-in) period begins from the **"Start"** date, followed by the respective starts of the training, validation, and testing periods. All dates are given as 00:00 UTC on January 1$^{\text{st}}$, unless noted (* refers to 00:00 UTC on January 17$^{\text{th}}$).

| Catalog | $\mathbf{M_c}$ | Start-Train-Val-Test-End | Train Events | Test Events |
|---|---|---|---|---|
| ETAS | 2.5 | 1971-1981-1998-2007-2020* | 117,550 | 43,327 |
| ETAS_incomplete | 2.5 | 1971-1981-1998-2007-2020* | 115,115 | 42,932 |
| Japan_Deprecated | 2.5 | 1990-1992-2007-2011-2020 | 22,213 | 15,368 |

are hampered when numerous events occur in quick succession, e.g. when phase arrivals of different events overlap at different stations or the amplitudes of small events are swamped by those of large events. Since catalog incompleteness increases for lower magnitude events, typically the task is to find the value $M_c$ above which there is approximately 100% detection probability. Choosing a truncation threshold $M_{\text{cut}}$ that is too high removes usable data. Where NPPs have demonstrated an ability to perform well with incomplete data (Stockman et al., 2023), typically a threshold below the completeness biases classical models such as ETAS (Seif et al., 2017). Seismologists often investigate the biases of different magnitude thresholds by performing repeat forecasting experiments for different thresholds (e.g. Mancini et al., 2022; Stockman et al., 2023), which we also facilitate in our datasets.

Typically $M_c$ is determined by comparing the raw earthquake catalog to the Gutenberg-Richter law (Gutenberg & Richter, 1936), which states that the distribution of earthquake magnitudes follows an exponential probability density function

$$f_{GR}(m) = \beta e^{-\beta(m-M_c)} \quad : m \geq M_c. \tag{8}$$

where $\beta$ is a rate parameter related to the b-value by $\beta = b \log 10$. Histogram-based approaches, such as the simple Maximum Curvature method (Wiemer & Wyss, 2000) as well as many others (e.g. Herrmann & Marzocchi, 2021, and references therein), identify the magnitude at which the observed catalog deviates from this law, indicating incompleteness (See Figure 6b).

In practice, catalog completeness varies in both time and space $M_c(t, \mathbf{x})$ (e.g. Schorlemmer & Woessner, 2008). During aftershock sequences, $M_c(t)$ can be very high (e.g., Agnew, 2015; Hainzl, 2016b) (See Figure 6a). Thresholding at the maximum value might remove too much data. Instead, modelers either omit particularly incomplete periods during training and testing (Kagan, 1991; Hainzl et al., 2008), model the incompleteness itself (Helmstetter et al., 2006; Werner et al., 2011; Omi et al., 2014; Hainzl, 2016a;b; Mizrahi et al., 2021; Hainzl, 2022), or accept known biases from disregarding this issue (Sornette & Werner, 2005). Spatially, catalogs are less complete farther from the seismic network (Mignan et al., 2011), so the spatial region can be constrained to remove outer, more incomplete areas (See Figure 6c).

## B  Additional Datasets

Beyond the official EarthquakeNPP datasets, we include 3 further datasets that either provide additional scientific insight or continuity from previous benchmarking works.

### B.1  Synthetic ETAS Catalogs.

We simulate a synthetic catalog using the ETAS model with parameters estimated from ComCat, at $M_c$ 2.5, within the same California region. A second catalog emulates the time-varying data-missingness present in observational catalogs by removing events using the time-dependent formula from Page et al. (2016),

$$M_c(M, t) = M/2 - 0.25 - \log_{10}(t), \tag{9}$$

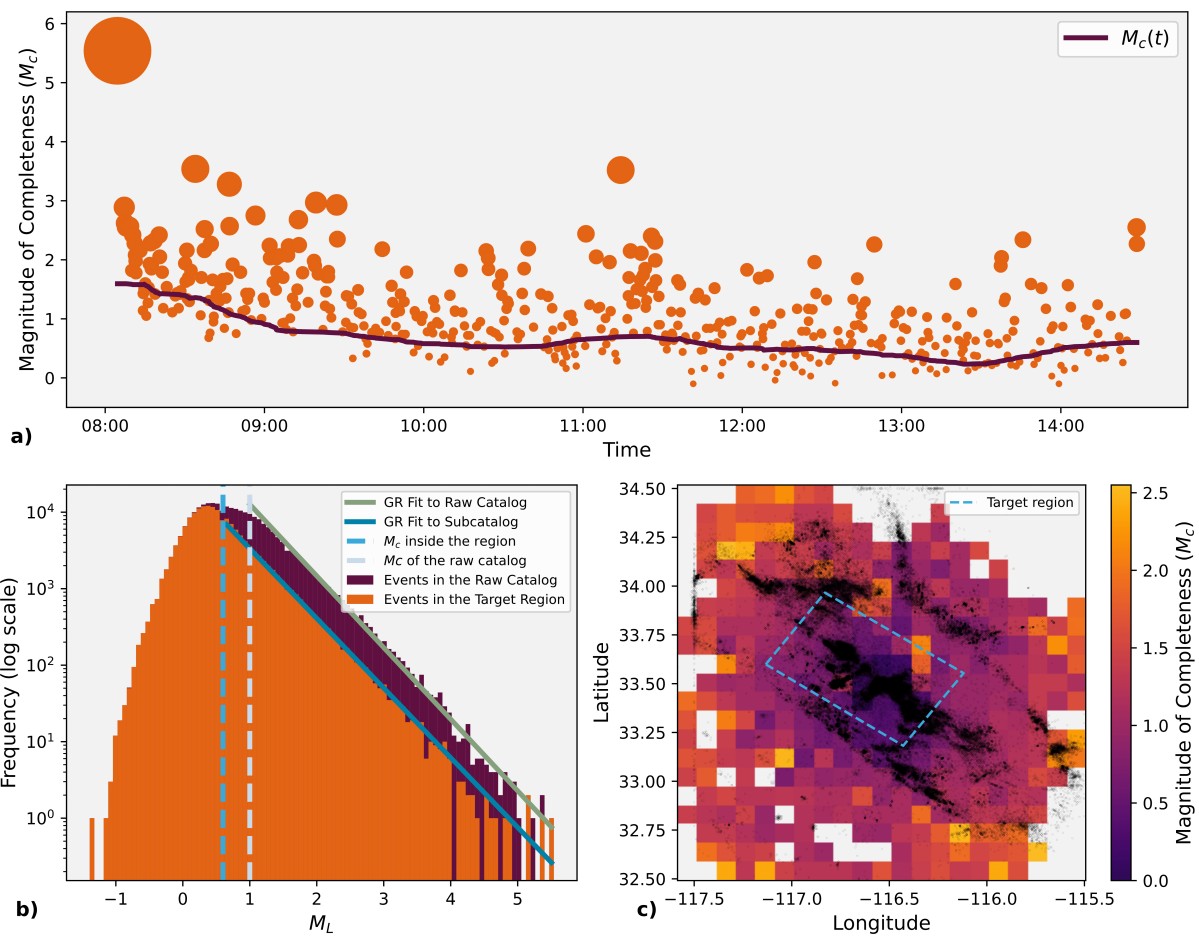

Figure 6: **a)** the June 10, 2016 Mw5.2 Borrego Springs earthquake and aftershocks, which occurred on the San Jacinto fault zone and is recorded in the WHITE catalog. An estimate of the magnitude of completeness $M_c(t)$ over time using the Maximum Curvature method reveals more incompleteness immediately following the large earthquake. **b)** magnitude-frequency histograms reveal that truncating the raw WHITE catalog to inside the target region decreases $M_c$. Each histogram is fit to the Gutenberg-Richter (GR) law and an estimate of $M_c$ for each catalog occurs where the histogram deviates from the (GR) line. **c)** An estimate of $M_c$ for gridded regions of the San Jacinto fault zone, using the raw WHITE catalog.

where $M$ is the mainshock magnitude. Events below this threshold are removed using mainshocks of Mw 5.2 and above. The inclusion of these datasets allows us to test whether NPPs are inhibited by data missingness to the same extent that ETAS is.

## B.2 Deprecated Catalog of Japan.

To provide continuity from the previous benchmarking for NPPs on earthquakes, we also provide results on the Japanese dataset from Chen et al. (2021), however with a chronological train-test split and without removing any supposed outlier events. To reflect our recommendation not to use this dataset in any future benchmarking following the dataset completeness issues mentioned above, we name this dataset `Japan_Deprecated`.

We can use this corrected dataset to quantify the inflation of performance caused by the non-chronological training-validation-testing splitting in the Chen et al. (2021) dataset. Table 4 presents the information gain (difference in total log-likelihood, see section 2.1) relative to a Poisson process for the three NPP models

Table 4: NPP performance comparison on the Original Chen et al. (2021) dataset versus the `Japan_Deprecated` dataset. Values are reported in terms of information gain from a homogeneous Poisson process.

| Model | Original Chen et al. (2021) dataset | Japan_Deprecated |
|---|---|---|
| NSTPP | 11.26 | 1.99 |
| DeepSTPP | 11.77 | 2.95 |
| AutoSTPP | 12.16 | 2.76 |

across the two datasets. The dramatic drop in information gain highlights how the original data split and omission of the 2011 Tohoku earthquake inflates model performance.

### B.3 Likelihood Evaluation

Figures 7 and 8 report the temporal and spatial log-likelihood scores of all the benchmarked models on additional datasets. On synthetic data generated by the ETAS model the performance of NPPs mirrors the results on the observational data (Figures 2 and 3). The performance of NPPs is more comparable to ETAS in terms of temporal log-likelihood however they cannot capture the distribution of earthquake locations. Change in temporal performance of models between the `ETAS` and `ETAS_incomplete` datasets reveal each model's robustness to the missing data typically present in earthquake catalogs (See section A.2). Auto-STPP and ETAS reduce in performance upon the removal earthquakes during aftershock sequences, whereas DeepSTPP and NSTPP maintain the same performance indicating a robustness to the data missingness.

On the `Japan_Deprecated` dataset, whilst ETAS remains the best performing model for spatial prediction, for temporal prediction it performs comparably to NSTPP and is even marginally outperformed by DeepSTPP. This performance can be attributed to the data completeness issues of the `Japan_Deprecated` dataset (see section 1.1), where the test period is missing all earthquakes bellow magnitude 4.0.

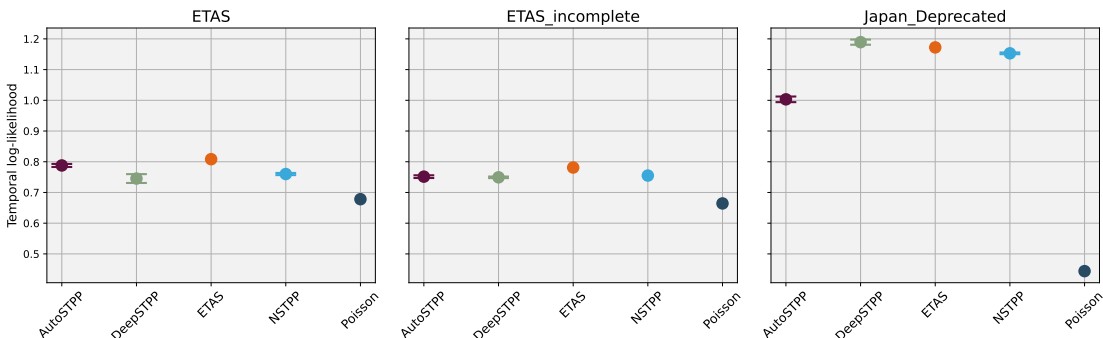

Figure 7: Test temporal log-likelihood scores for all the spatio-temporal point process models on each of the additional datasets. Error bars of the mean and standard deviation are constructed for the NPPs using three repeat runs.

## C Effect of Training Window on ETAS Performance

To verify that fitting ETAS on both the training and validation windows does not artificially improve its performance relative to the NPPs, we retrained ETAS using only the training window and re-evaluated its

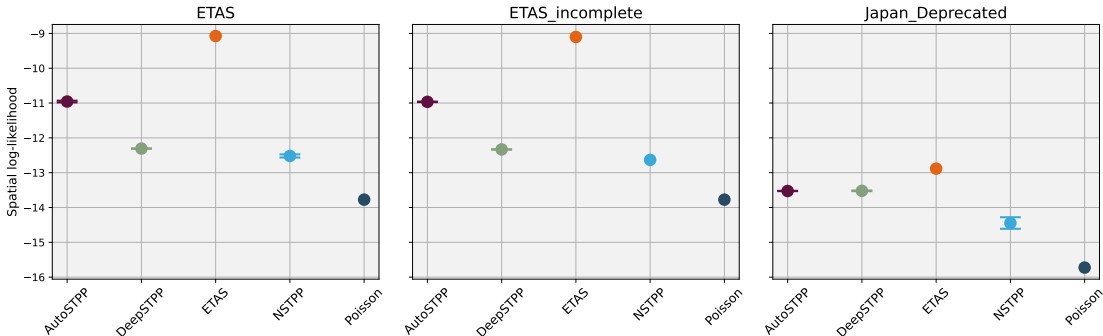

Figure 8: Test spatial log-likelihood scores for all the spatio-temporal point process models on each of the additional datasets. Error bars of the mean and standard deviation are constructed for the NPPs using three repeat runs.

test log-likelihoods. As shown in Figures 9 & 10, the resulting log-likelihood scores are effectively unchanged across all EarthquakeNPP datasets.

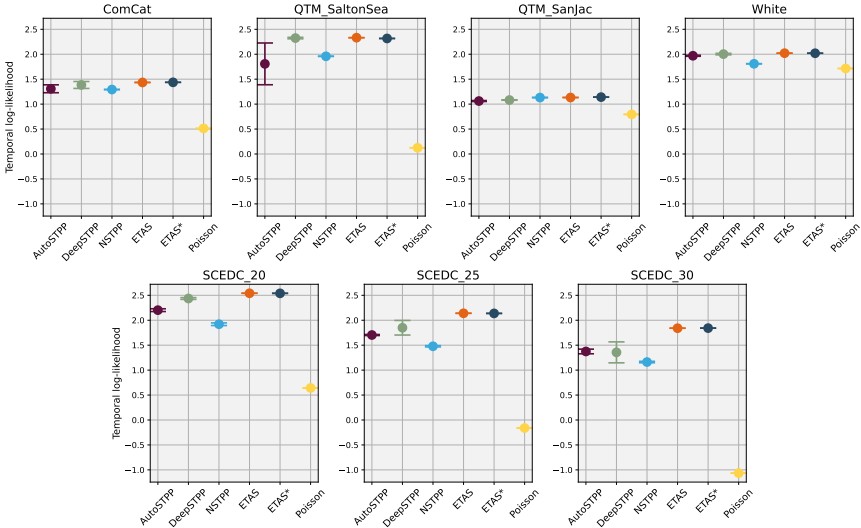

Figure 9: Test temporal log-likelihood scores for all the spatio-temporal point process models on each of the EarthquakeNPP datasets. Error bars of the mean and standard deviation are constructed for the NPPs using three repeat runs. `ETAS` (orange) is trained on both training and validation windows, whereas `ETAS*` (light blue) is trained only using the training window.

# D  Computational Efficiency

## D.1  Training

Table 5 reports the training times for each model across all datasets. We ran all the NPP models using a HPC node with Nvidia Ampere GPU with 4x Nvidia A100 40GB SXM "Ampere" GPUs and AMD EPYC 7543P 32-Core Processor "Milan" CPU using torch==1.12.0 and cuda==11.3.

ETAS training scales $\mathcal{O}(n^2)$ with the total number of events, since for every event a contribution to the intensity function is computed from a summation over all previous events. This scaling, coupled with the

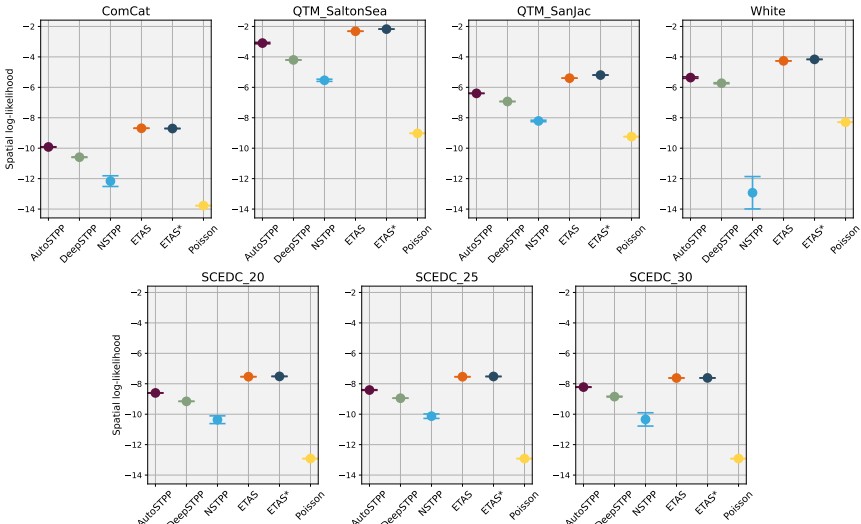

Figure 10: Test spatial log-likelihood scores for all the spatio-temporal point process models on each of the EarthquakeNPP datasets. Error bars of the mean and standard deviation are constructed for the NPPs using three repeat runs. `ETAS` (orange) is trained on both training and validation windows, whereas `ETAS*` (light blue) is trained only using the training window.

Table 5: Training times for each model across all datasets, including the number of training events. Times are formatted as HH:MM:SS, with days included for durations exceeding 24 hours. SMASH times are estimated as $1.5\times$ AutoSTPP, and DSTPP times are extrapolated assuming linear scaling from Salton Sea.

| Dataset | # Training Events | ETAS | Deep-STPP | AutoSTPP | NSTPP | SMASH | DSTPP | Poisson |
|---|---|---|---|---|---|---|---|---|
| ComCat | 79,037 | 08:59:04 | 00:15:35 | 01:34:09 | 3 days, 05:10:17 | 2:21:13 | 20:05:57 | <1 second |
| QTM_SaltonSea | 44,042 | 07:28:28 | 00:26:46 | 01:45:34 | 2 days, 00:26:45 | 2:38:21 | 11:12:00 | <1 second |
| QTM_SanJac | 18,664 | 00:32:40 | 00:09:31 | 00:37:03 | 1 day, 22:06:33 | 0:55:34 | 4:44:46 | <1 second |
| SCEDC_20 | 128,265 | 13:42:30 | 00:38:10 | 02:54:51 | 3 days, 02:20:40 | 4:22:16 | 1 day, 8:37:05 | <1 second |
| SCEDC_25 | 43,221 | 03:09:14 | 00:09:34 | 00:56:05 | 2 days, 16:33:55 | 1:24:07 | 10:59:28 | <1 second |
| SCEDC_30 | 12,426 | 00:42:25 | 00:02:44 | 00:16:01 | 1 day, 16:39:04 | 0:24:01 | 3:10:26 | <1 second |
| White | 38,556 | 03:55:40 | 00:08:21 | 01:10:51 | 2 days, 01:03:57 | 1:46:17 | 9:48:47 | <1 second |
| Japan_Deprecated | 22,213 | 06:09:08 | 00:13:45 | 01:02:07 | 2 days, 05:32:03 | 1:33:06 | 5:39:32 | <1 second |
| ETAS | 117,550 | 00:33:25 | 00:15:24 | 01:10:22 | 3 days, 03:09:17 | 1:45:33 | 1 day, 1:27:44 | <1 second |
| ETAS_incomplete | 115,115 | 00:35:14 | 00:15:29 | 01:09:43 | 3 days, 11:39:51 | 1:44:34 | 1 day, 2:28:42 | <1 second |

lack of parallelization in the current implementation, results in long training times for larger datasets. Poorer scaling will likely hinder ETAS if dataset sizes continue to grow in the future (Stockman et al., 2024).

Encouragingly, both DeepSTPP and AutoSTPP are significantly faster to train due to GPU acceleration and their use of a sliding window of the most recent $k = 20$ events. While exact complexity analyses are not provided in Zhou et al. (2022) or Zhou & Yu (2024), we can infer that DeepSTPP likely scales as $\mathcal{O}(kn)$ since it benefits from a closed-form expression for the likelihood. AutoSTPP, though requiring automatic

integration to compute the likelihood, still scales with $\mathcal{O}(kn)$ because the additional integration cost does not affect the overall scaling.

NSTPP, on the other hand, incurs significant training costs, rendering it impractical for real-time forecasting. Unlike the sliding window mechanism used in DeepSTPP and AutoSTPP, NSTPP partitions the event sequence into fixed time intervals, leading to sequences that are much longer than the $k = 20$ events used by the other models (as shown in Figure 11 of Chen et al. (2021)). Furthermore, solving an ODE for each event time adds a significant computational burden, even with the use of their faster attentive CNF architecture.

Whilst SMASH and DSTPP are built on the same backbone architecture, SMASH is much quicker to train than DSTPP, even faster than ETAS. This efficiency arises from its use of a single-step, normalization-free score-matching objective, which avoids the costly denoising and sampling loops required in diffusion-based training. SMASH directly learns the gradient of the log-density via pseudolikelihood estimation, enabling efficient GPU parallelization and bypassing the need for repeated evaluations over diffusion steps. In contrast, DSTPP simulates a sequential generative process over hundreds of intermediate steps per sample, significantly increasing computation and memory costs.

## D.2    Inference

Whilst log-likelihood computation for ETAS, Poisson, DeepSTPP, AutoSTPP, and NSTPP is fast ($< 30$ seconds per dataset), real-time earthquake forecasting and CSEP evaluation require simulating many repeated event sequences (at least 10,000) over a fixed forecasting horizon. While ETAS training scales as $\mathcal{O}(n^2)$ with the number of training events, its simulation is considerably more efficient, scaling approximately as $\mathcal{O}(n \log n)$ due to its equivalent formulation as a Hawkes branching process (see Section 2.2), which enables immigration–birth sampling.

Fast simulation is not currently feasible for several NPP architectures. NSTPP is not Hawkes-based and requires solving a neural ordinary differential equation to generate each new event, rendering simulation prohibitively slow even for small datasets. DeepSTPP and AutoSTPP, while inspired by Hawkes processes, employ non-stationary neural triggering kernels that depend on the full event history. As a result, standard immigration–birth sampling cannot be applied, since triggering relationships change after each event and new events cannot be generated independently or in parallel. Simulation via thinning is also problematic: AutoSTPP does not enforce monotonic or decaying kernels, meaning the conditional intensity $\lambda^*(t, \mathbf{x})$ cannot be safely upper-bounded, while DeepSTPP can in principle be simulated via thinning but is extremely slow in practice. For these reasons, DeepSTPP, AutoSTPP, and NSTPP are excluded from CSEP generative evaluation. For the models where large-scale simulation is tractable (ETAS, SMASH, and DSTPP), we report inference and simulation times in Table 6. Based on simulation times, DSTPP is fastest, followed by ETAS, with SMASH consistently the slowest across datasets, reflecting the fact that DSTPP samples events via a fixed-length diffusion process with closed-form updates, whereas SMASH relies on iterative Langevin dynamics requiring many gradient evaluations per event.

Table 6: Simulation time for a batch of 100 repeated simulations across datasets. Reported times correspond to individual forecast days and are summarised as minimum, median, mean, and maximum over the testing window, reflecting variation in daily event counts rather than runtime stochasticity. Times are formatted as HH:MM:SS.

| Dataset | No. Days | Daily Counts | ETAS | SMASH | DSTPP |
|---|---|---|---|---|---|
| | | (min / med / mean / max) | (min / med / mean / max) | (min / med / mean / max) | (min / med / mean / max) |
| ComCat | 4764 | 0 / 3 / 4.59 / 1241 | 00:00:24 / 00:01:25 / 00:01:33 / 01:49:25 | 00:02:40 / 00:03:41 / 00:04:20 / 08:26:00 | 00:00:08 / 00:01:10 / 00:01:12 / 00:31:20 |
| SaltonSea | 731 | 0 / 2 / 5.61 / 292 | 00:00:19 / 00:01:20 / 00:01:39 / 00:26:38 | 00:02:15 / 00:03:17 / 00:04:45 / 02:00:57 | 00:00:06 / 00:01:08 / 00:01:14 / 00:08:12 |
| SanJac | 731 | 0 / 5 / 6.02 / 241 | 00:00:34 / 00:01:35 / 00:01:41 / 00:22:11 | 00:03:28 / 00:04:30 / 00:04:55 / 01:40:15 | 00:00:12 / 00:01:13 / 00:01:14 / 00:06:58 |
| SCEDC | 2191 | 0 / 3 / 5.96 / 2134 | 00:00:23 / 00:01:25 / 00:01:40 / 03:07:19 | 00:02:40 / 00:03:41 / 00:04:54 / 14:28:21 | 00:00:07 / 00:01:10 / 00:01:14 / 00:53:05 |
| WHITE | 1461 | 0 / 13 / 16.48 / 520 | 00:01:15 / 00:02:17 / 00:02:36 / 00:46:31 | 00:06:43 / 00:07:45 / 00:09:10 / 03:33:28 | 00:00:23 / 00:01:24 / 00:01:30 / 00:13:46 |

# E   Analysis of Likelihood Scores

## E.1   Temporal Information Gain

To better interpret the temporal likelihood results presented in Figure 2, we analyse how model performance evolves over time in each dataset in Figure 11, presenting the cumulative information gain (log-likelihood difference) of each NPP model over ETAS.

In Figure 11a on the `ComCat` dataset, the largest decreases in relative performance occur during the two largest sequences in the testing window, namely the 2010 El Mayor Cucapah M7.2 and the 2019 Ridgecrest M7.1 earthquakes. ETAS performs strongly in these periods, likely because it incorporates magnitude scaling which enables it to model large aftershock cascades effectively. In contrast, the 2014 South Napa M6.0 and 2014 Offshore Eureka M6.8 do not produce such a marked drop in relative performance. This suggests that capturing the largest aftershock sequences is a key limitation of current NPP models, while during background periods NPPs perform relatively better due to their ability to capture non-stationarity, a property directly not modelled by ETAS. Despite the overall worse performance of NSTPP, the relative decrease during large events is not as sharp as the other models, suggesting superior performance.

A similar pattern appears in Figure 11b for the `SCEDC` dataset. All models show a sharp reduction in performance during the 2019 Ridgecrest M7.1 sequence. During the quieter period leading up to Ridgecrest, DeepSTPP performs comparably to ETAS. This again suggests that the lack of explicit magnitude conditioning limits NPP performance during large sequences and highlights NPP models ability to capture background non-staionarities.

Figures 11c, 11d and 11e show results for the smaller regional datasets, `SanJac`, `SaltonSea` and `White`. These smaller magnitude, more regionally concentrated datasets, don't display the same constrasting "background", "mainshock" behaviour present in `ComCat` and `SCEDC`. Although the overall performance of the NPP models is below that of ETAS, improvements occur during the 2017 Brawley swarm in the `SaltonSea` dataset, whereby DeepSTPP achieves higher temporal likelihood than ETAS. This behaviour aligns with the known difficulty ETAS has in modelling swarm-like sequences that are not initiated by a large mainshock.

## E.2   Spatial Information Gain

To better interpret the spatial likelihood results presented in Figure 3, we analyse how model performance evolves over space in each dataset in Figures 12, 13, visualising geographically where NPPs outperform or fall behind ETAS.

Figures 12 and 13 show the spatial distribution of log-likelihood information gain across all datasets. Across both figures, a consistent pattern emerges. ETAS performs best in regions dominated by large, magnitude driven mainshock–aftershock sequences, while NPP performance degrades in the immediate vicinity of these events. In contrast, NPPs tend to perform more competitively in regions characterised by spatially complex or diffuse seismicity.

In the larger regional catalogs shown in Figure 12, this distinction is most apparent in the `ComCat` dataset. Near the Ridgecrest and El Mayor–Cucapah sequences, NPPs exhibit reduced information gain relative to ETAS, consistent with the absence of explicit magnitude driven triggering. However, in the complex tectonic setting of the Mendocino Triple Junction, NPPs achieve comparatively strong spatial performance, with frequent positive information gain relative to ETAS. This region is characterised by interacting fault systems and persistent background activity rather than a single dominant mainshock, suggesting that NPPs are better suited to modelling such non-stationary and spatially heterogeneous seismic regimes.

Figure 13 presents results for the smaller regional datasets San Jacinto, Salton Sea, and White. These regions are dominated by lower magnitude seismicity and swarm like behaviour, and here NPPs again perform more competitively relative to ETAS. AutoSTPP assigns probability more smoothly along fault structures similarly to ETAS, resulting in information gain values concentrated near zero, while DeepSTPP and NSTPP show more heterogeneous behaviour. NSTPP produces more extreme and spikier information gain values that are spatially scattered, indicating less precise spatial localisation despite occasional high

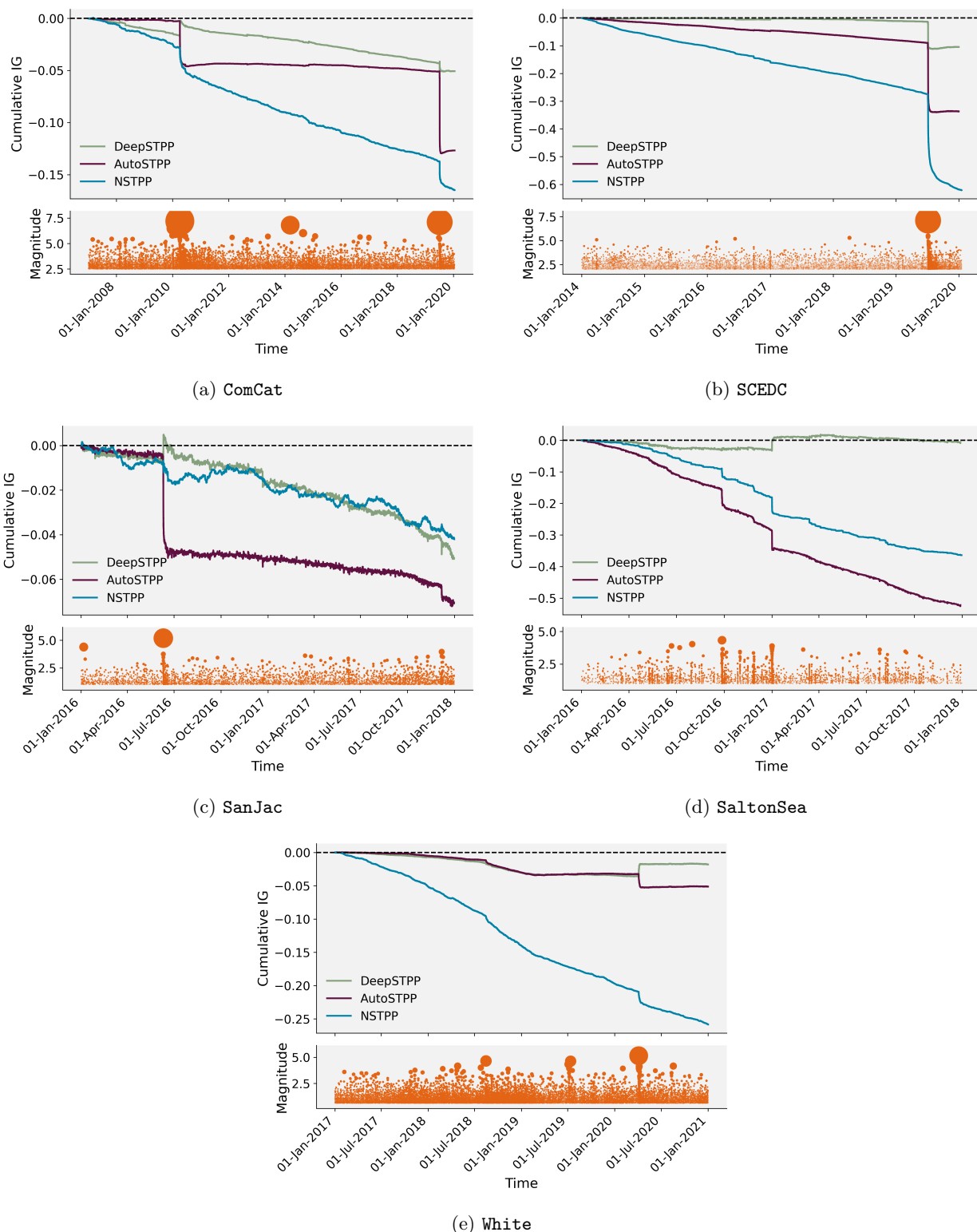

Figure 11: Cumulative information gain (IG) plots for the temporal performance of all the NPP models with respect to ETAS on a) `ComCat`,b) `SCEDC`, c) `QTM_San_Jac`, d) `QTM_Salton_Sea`, e) `White`.

likelihood assignments. Instability during training led to the drastic underperformance of NSTPP on the `White` dataset and consequent low likelihood scores distributed across the entire region.

Overall, the spatial results mirror the temporal analysis. Current NPP architectures struggle most in mainshock dominated regimes but show clear promise in modelling spatially complex background seismicity and swarm driven activity, motivating future work on incorporating large magnitude triggering while preserving this flexibility.

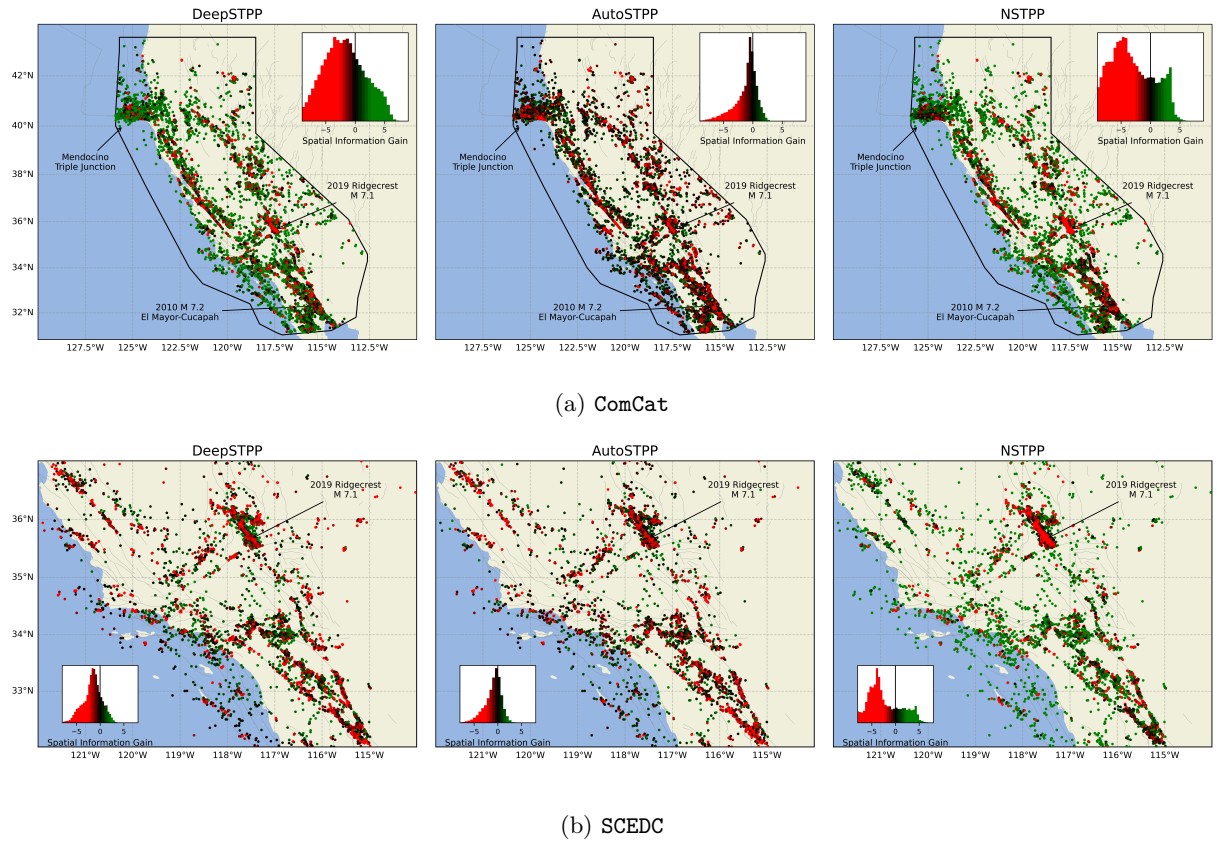

(a) `ComCat`

(b) `SCEDC`

Figure 12: Spatial information gain per event, for NPP models relative to ETAS on (a) `ComCat` and (b) `SCEDC`. Scatter points correspond to the geographical location of the forecasted event, coloured by the value of the information gain over the ETAS model (green positive, red negative). For each model an inset plot displays the distribution of the spatial information gains for all events in the testing period.

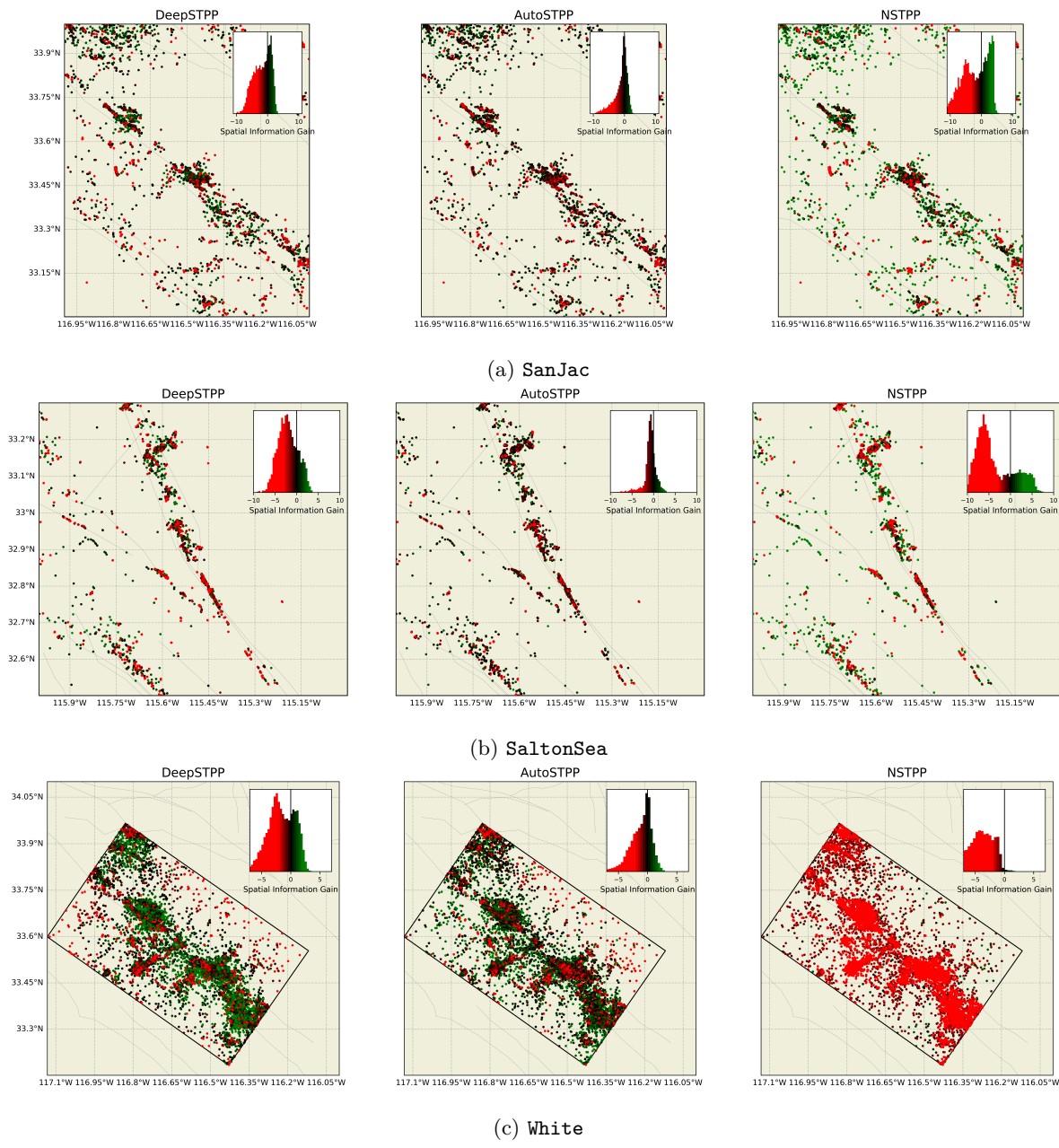

Figure 13: Spatial information gain per event, for NPP models relative to ETAS on (a) `SanJac`, (b) `SaltonSea`, and (c) `White`. Scatter points correspond to the geographical location of the forecasted event, coloured by the value of the information gain over the ETAS model (green positive, red negative). For each model an inset plot displays the distribution of the spatial information gains for all events in the testing period.

## F  CSEP Consistency Tests

### F.1  Number (Temporal) Test

The number test evaluates the temporal component of the forecast by checking the consistency of the forecasted number of events, $N$ with those observed in the forecast horizon, $N_{\text{obs}}$. Upper and lower quantiles

are estimated using the empirical cumulative distribution from the repeat simulations, $F_N$,

$$\delta_1 = \mathbb{P}(N \geq N_{\text{obs}}) = 1 - F_N(N_{\text{obs}} - 1) \tag{10}$$
$$\delta_2 = \mathbb{P}(N \leq N_{\text{obs}}) = F_N(N_{\text{obs}}). \tag{11}$$

### F.2  Pseudo-Likelihood Test

The pseudo-likelihood test evaluates the compatibility of a forecast with an observed catalog using an approximation to the space-time point process likelihood.

The test statistic is based on the pseudo-log-likelihood:

$$\hat{L}_{\text{obs}} = \sum_{i=1}^{N_{\text{obs}}} \log \hat{\lambda}_s(k_i) - \bar{N}, \tag{12}$$

where $\hat{\lambda}_s(k_i)$ is the approximate rate density in the spatial cell of the $i^{\text{th}}$ event, and $\bar{N}$ is the expected number of events.

Each forecast simulation $j$ provides a test statistic

$$\hat{L}_j = \sum_{i=1}^{N_j} \log \hat{\lambda}_s(k_{ij}) - \bar{N}, \tag{13}$$

which is used to build the empirical cumulative distribution $F_L$. The quantile score is then computed as

$$\gamma_L = \mathbb{P}(\hat{L}_j \leq \hat{L}_{\text{obs}}) = F_L(\hat{L}_{\text{obs}}). \tag{14}$$

### F.3  Spatial Test

To evaluate the spatial component of the forecast, a test statistic aggregates the forecasted rates of earthquakes over a regular grid,

$$S = \left[ \sum_{i=1}^{N} \log \hat{\lambda}(k_i) \right] N^{-1}, \tag{15}$$

where $\hat{\lambda}(k_i)$ is the approximate rate in the cell $k$ where the $i^{th}$ event is located. Upper and lower quantiles are estimated by comparing the observed statistic

$$S_{\text{obs}} = \left[ \sum_{i=1}^{N_{\text{obs}}} \log \hat{\lambda}(k_i) \right] N_{\text{obs}}^{-1}, \tag{16}$$

with the empirical cumulative distribution of $S$ using the repeat simulations, $F_S$

$$\gamma_s = \mathbb{P}(S \leq S_{\text{obs}}) = F_S(S_{\text{obs}}). \tag{17}$$

The grid is constructed from $\{0.1°, 0.05°, 0.01°\}$ squares for `ComCat`, `SCEDC` and $\{$`QTM_Salton_Sea`, `QTM_SanJac`, `White`$\}$ respectively.

### F.4  Magnitude Test

To evaluate the earthquake magnitude component of the forecast, a test statistic compares the histogram of a forecast's magnitudes $\Lambda^{(m)}$, against the mean histogram over all forecasts $\bar{\Lambda}^{(m)}$,

$$D = \sum_k \left( \log \left[ \bar{\Lambda}^{(m)}(k) + 1 \right] - \log \left[ \Lambda^{(m)}(k) + 1 \right] \right)^2, \tag{18}$$

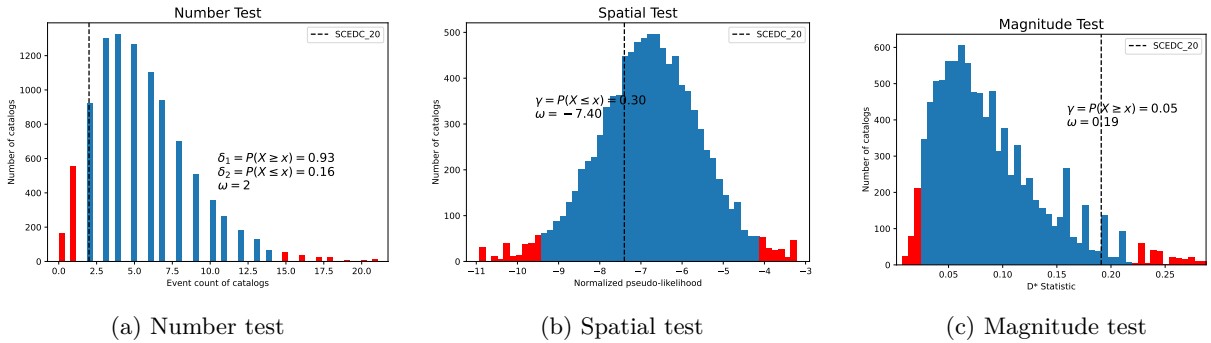

(a) Number test  (b) Spatial test  (c) Magnitude test

Figure 14: CSEP consistency tests on the ETAS model for the first day (01/01/2014) of the testing period in the `SCEDC` catalog. A total of 10,000 simulations are generated to compute empirical distributions of the test statistics for each of the three consistency tests: (a) Number test, (b) Spatial test, and (c) Magnitude test. The test fails if the observed statistic falls within the rejection region (red), defined by the 0.05 and 0.95 quantiles of the distribution.

where $\Lambda^{(m)}(k)$ and $\bar{\Lambda}^{(m)}(k)$ are the counts in the $k^{th}$ bin of the forecast and mean histograms, normalised to have the same total counts as the observed catalog. Upper and lower quantiles are estimated by comparing the observed statistic

$$D_{\text{obs}} = \sum_k \left( \log \left[ \bar{\Lambda}^{(m)}(k) + 1 \right] - \log \left[ \Lambda_{\text{obs}}^{(m)}(k) + 1 \right] \right)^2, \tag{19}$$

with the empirical distribution of $D$ using the repeat simulations, $F_D$

$$\gamma_m = \mathbb{P}(D \leq D_{\text{obs}}) = F_D(D_{\text{obs}}). \tag{20}$$

Histogram bins of size $\delta_m = 0.1$ are used across all datasets.

Although SMASH is, in principle, capable of modelling earthquake magnitudes, we restrict it to rate forecasting in this benchmark, as extending it to fine-grained magnitude prediction led to a deterioration in spatio-temporal performance metrics during training.

### F.5 Evaluating Multiple Forecasting Periods

Savran et al. (2020) describe how to assess a model's performance across the multiple days in the testing period (Figure 15). By construction, quantile scores over multiple periods should be uniformly distributed if the model is the data generator (Gneiting & Katzfuss, 2014). Therefore comparing quantile scores against standard uniform quantiles (y = x), highlights discrepancies between the observed data and the forecast. Additional statements can be made about over-prediction or under-prediction of each test statistic (quantile curves above/bellow y=x respectively). The Kolmogorov-Smirnov (KS) statistic then quantifies the degree of difference to the uniform distribution for each of the tests.

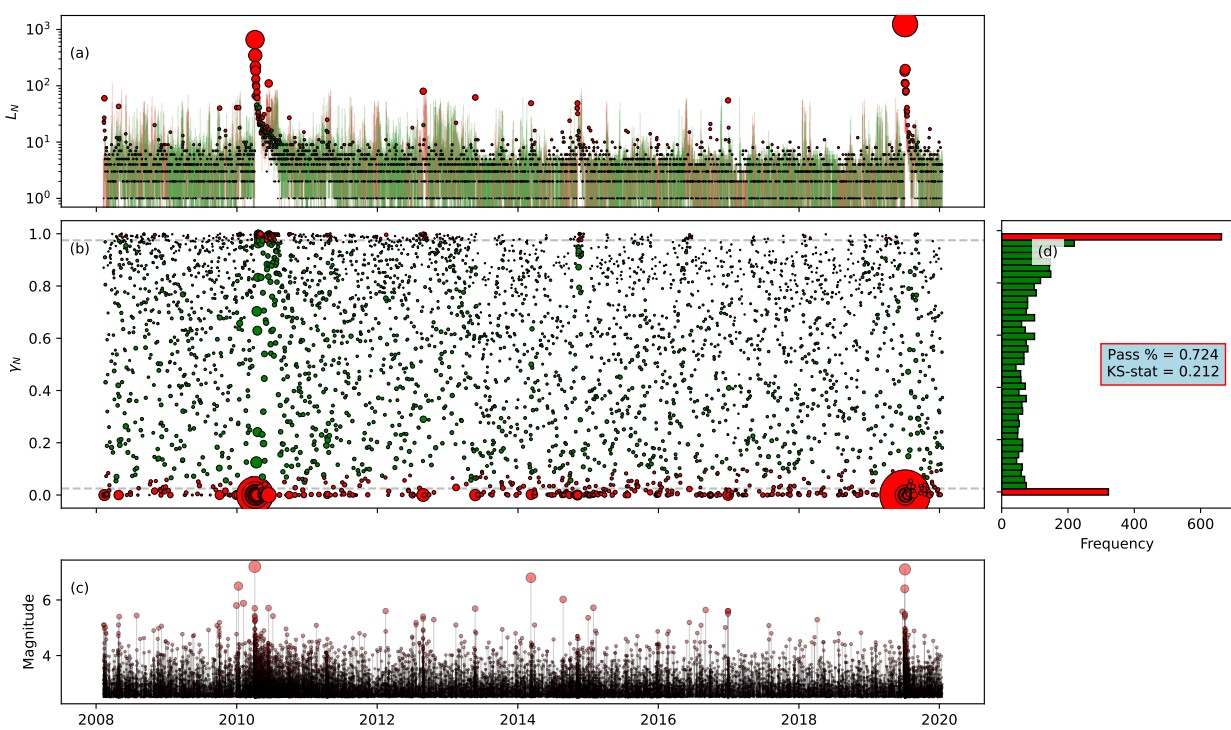

Figure 15: Daily number forecasts from SMASH on the `ComCat` dataset. (a) Forecasted daily distributions for the number of earthquakes, with green lines indicating days where the observed count falls within the 95% forecast interval, and red lines where the forecast fails. Observed values are marked with dot sizes proportional to the number of earthquakes. (b) Quantile scores from the number test for each day, with red markers indicating failed forecasts. Marker size reflects the number of earthquakes observed on that day. (c) Temporal evolution of observed earthquakes during the testing period, with event magnitudes represented by marker size. (d) Histogram of quantile scores from the number test. Under ideal calibration, scores should follow a uniform distribution. Red bars indicate failed forecasts, and the Kolmogorov–Smirnov (KS) statistic quantifies deviation from uniformity.

# G  Analysis of CSEP Tests

## G.1  Temporal

To further interpret the CSEP consistency test results reported in Table 2, Figures 16–18 show the daily event count forecasts produced by ETAS, SMASH, and DSTPP across all EarthquakeNPP datasets. These plots reveal systematic differences in how the models capture both background seismicity and sudden increases in earthquake rate.

Across all datasets, ETAS provides the most consistent forecasts of daily event counts. It captures low-activity days well and responds more effectively than the NPP-based models to increases in seismicity rate, although it still underpredicts the largest rate excursions associated with major earthquake sequences. This behaviour is particularly evident in the ComCat and SCEDC datasets during the 2010 El Mayor–Cucapah and 2019 Ridgecrest sequences (Figures 16a and 16b), and explains ETAS's consistently high consistency test pass rates.

SMASH exhibits highly variable daily rate estimates across all regions. While this variability occasionally allows it to match periods of elevated seismicity, such as during El Mayor–Cucapah in ComCat, it more often leads to pronounced over and under prediction. This spiky behaviour results in a substantial number of failed consistency tests.

DSTPP produces much smoother daily rate forecasts than SMASH, but this comes at the cost of systematic underprediction. Across all datasets, DSTPP underestimates both background seismicity and elevated rate periods, with the effect becoming especially severe in SCEDC and White (Figures 16b and 18). This persistent bias explains its low consistency test pass rates.

Performance differences across datasets largely reflect the dominant seismic regime. In smaller regions such as San Jacinto and Salton Sea (Figures 17a and 17b), all models perform more competitively due to the absence of large mainshock-driven rate increases. However, even in these settings, SMASH remains overly variable and DSTPP continues to underestimate daily rates.

## G.2  Spatial

Figures 19 and 20 show aggregated spatial forecasts over the entire testing period for ETAS, SMASH, and DSTPP across all EarthquakeNPP datasets. These plots summarise how each model distributes seismicity rate spatially when forecasts are integrated over time. While this provides a useful overview of long-term spatial structure, it does not distinguish whether high rates are forecast *before* events occur.

Across all datasets, ETAS produces the most spatially concentrated forecasts, with high rates aligned along known fault structures and low rates assigned to seismically inactive regions. This apparent spatial precision arises largely from ETAS's explicit modelling of clustering, whereby elevated rates are assigned in the vicinity of earthquakes that have already occurred. This behaviour is particularly evident in the ComCat and SCEDC datasets (Figures 19a and 19b), where ETAS reproduces the large-scale fault network and major clusters associated with past seismicity.

In contrast, SMASH consistently produces spatially diffuse forecasts. While it sometimes captures regions of elevated activity, its rate is spread broadly across each domain, leading to weaker spatial contrast between active and inactive regions. This effect is most pronounced in ComCat, where SMASH concentrates strongly around the southern end of the domain near the 2010 El Mayor–Cucapah sequence, while remaining diffuse elsewhere. Similar behaviour is observed in SCEDC, where SMASH is again dominated by the El Mayor region and assigns comparatively high rates across large areas of the domain.

DSTPP generally produces smoother spatial forecasts than ETAS but with greater structure than SMASH. In ComCat, San Jacinto, and Salton Sea (Figures 19a, 20a, and 20b), DSTPP follows the main fault-aligned clustering while attenuating sharp spatial contrasts. However, in SCEDC and White, DSTPP substantially underestimates the overall seismicity rate, resulting in uniformly low spatial forecasts and visible boundary effects, particularly near the edges of the domains (Figures 19b and 20c).

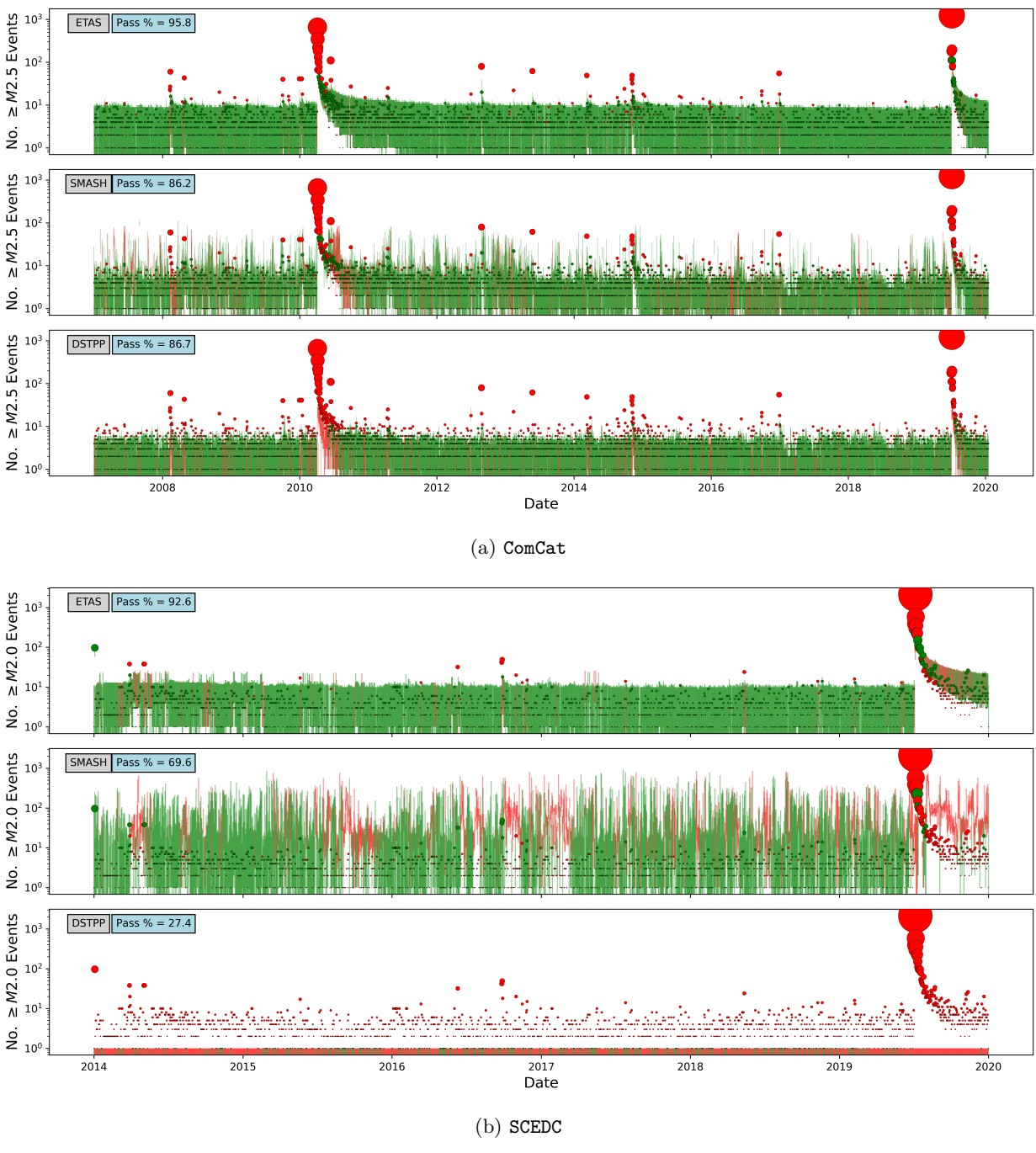

(a) ComCat

(b) SCEDC

Figure 16: Daily number forecasts from ETAS, SMASH, and DSTPP over the full testing period for (a) ComCat and (b) SCEDC. Vertical lines show the forecasted daily distributions of earthquake counts. Green lines indicate days where the observed count falls within the 95% forecast interval, while red lines indicate failures. Observed daily counts are shown as dots, with marker size proportional to the number of earthquakes and colour indicating pass (green) or fail (red).

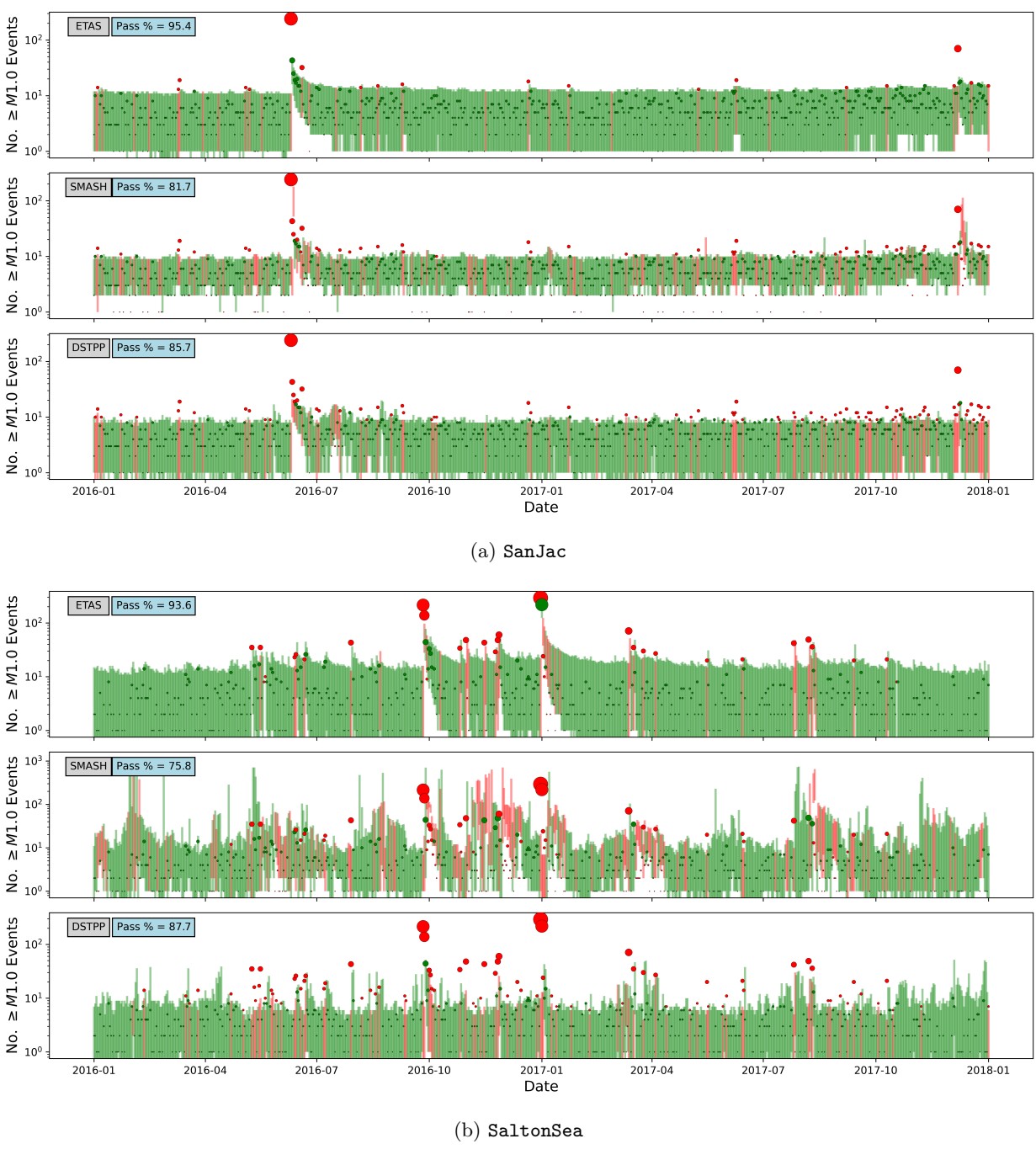

Figure 17: Daily number forecasts from ETAS, SMASH, and DSTPP over the full testing period for (a) `SanJac` and (b) `SaltonSea`. Vertical lines show the forecasted daily distributions of earthquake counts. Green lines indicate days where the observed count falls within the 95% forecast interval, while red lines indicate failures. Observed daily counts are shown as dots, with marker size proportional to the number of earthquakes and colour indicating pass (green) or fail (red).

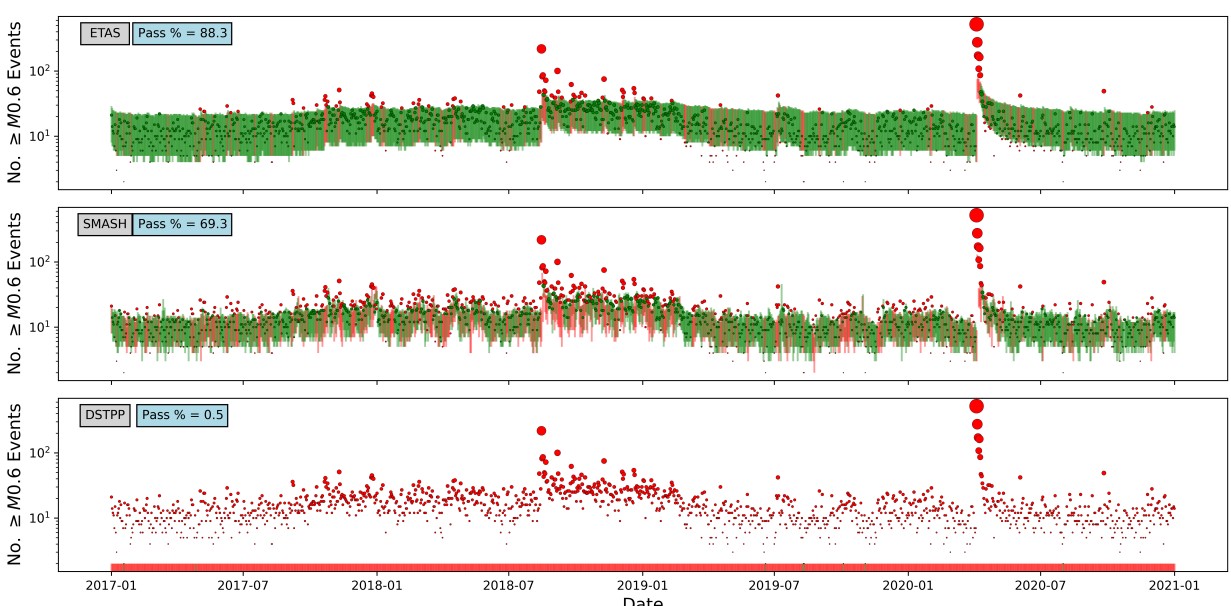

Figure 18: Daily number forecasts from ETAS, SMASH, and DSTPP over the full testing period for `White`. Vertical lines show the forecasted daily distributions of earthquake counts. Green lines indicate days where the observed count falls within the 95% forecast interval, while red lines indicate failures. Observed daily counts are shown as dots, with marker size proportional to the number of earthquakes and colour indicating pass (green) or fail (red).

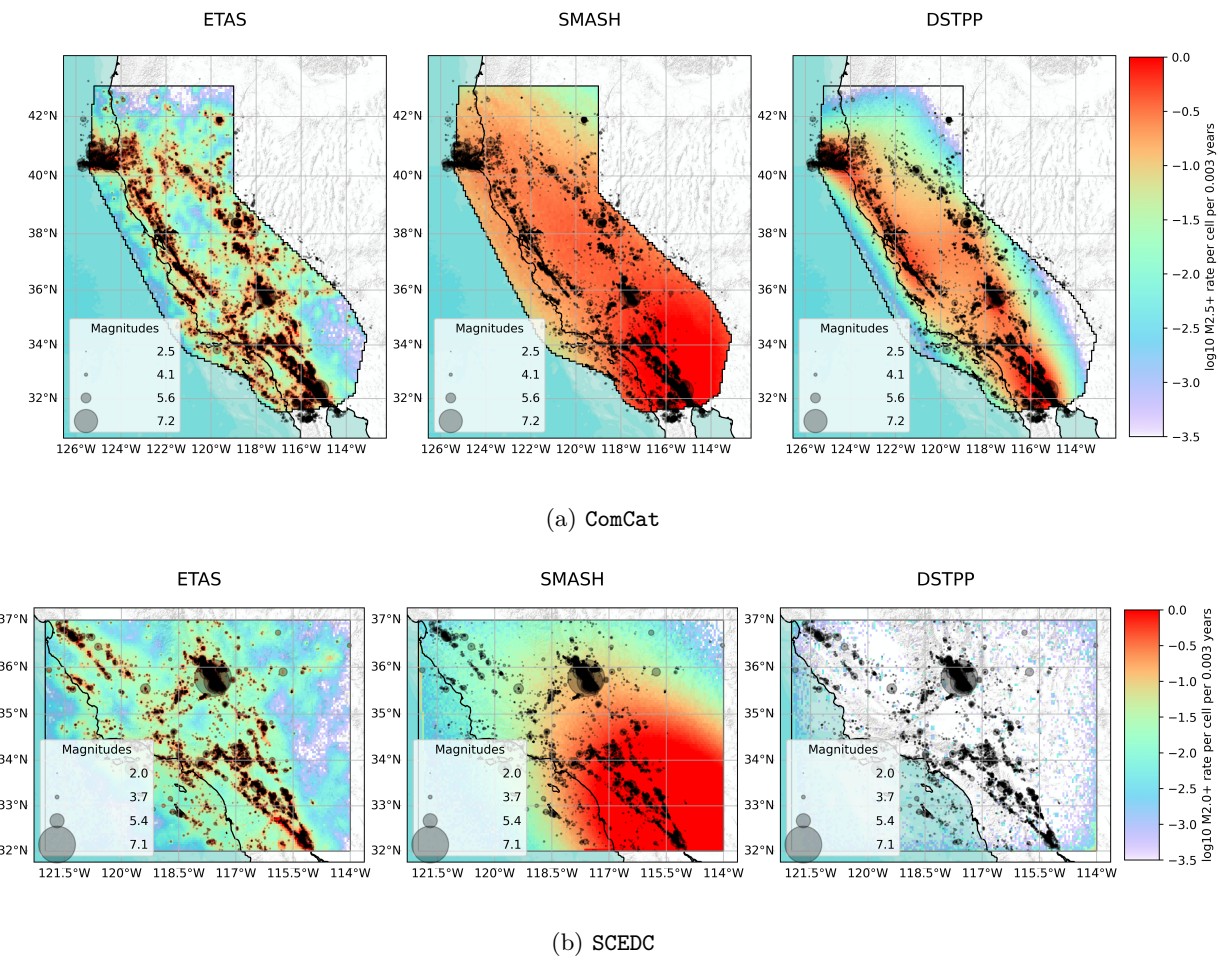

Figure 19: Spatial forecasts from ETAS, SMASH and DSTPP, aggregated across the entire testing period of (a) ComCat and (b) SCEDC. For each day in the testing period, every model simulates 10,000 repeated earthquake catalogs within the boundary region. All simulated catalogs are aggregated across the entire testing period and the expected earthquake rates are plotted within each grid cell. Observed earthquakes are overlaid with marker size proportional to magnitude.

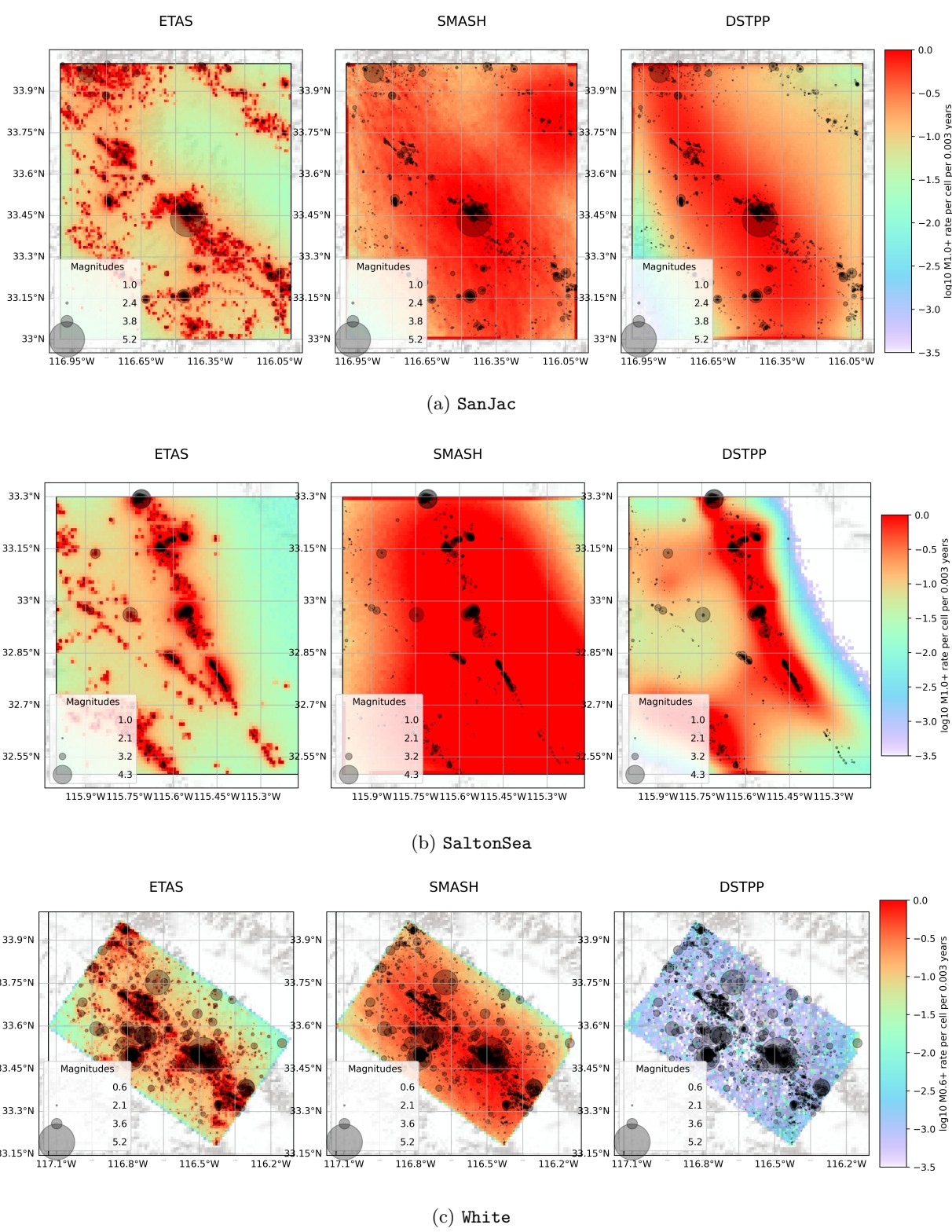

Figure 20: Spatial forecasts from ETAS, SMASH and DSTPP, aggregated across the entire testing period of (a) `SanJac`, (b) `SaltonSea`, and (c) `White`. For each day in the testing period, every model simulates 10,000 repeated earthquake catalogs within the boundary region. All simulated catalogs are aggregated across the entire testing period and the expected earthquake rates are plotted within each grid cell. Observed earthquakes are overlaid with marker size proportional to magnitude.

# H Further Dataset Figures

## H.1 `ComCat`

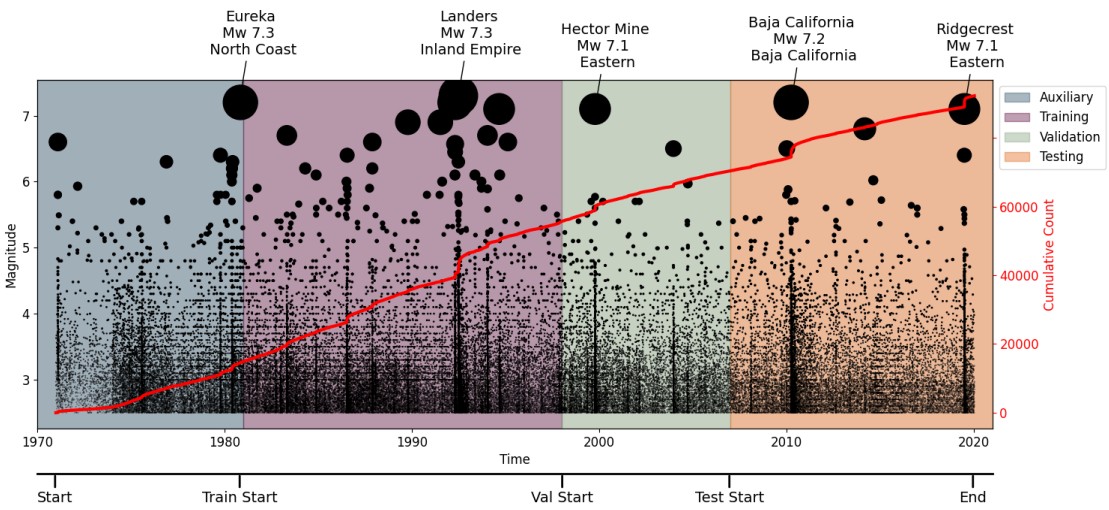

Figure 21: Times and magnitudes of events in the `ComCat` dataset (with key events labeled). The size of the points are plotted on a log scale corresponding to Mw. Auxiliary, training, validation and testing periods are indicated by colour and a further cumulative count of events is indicated in red.

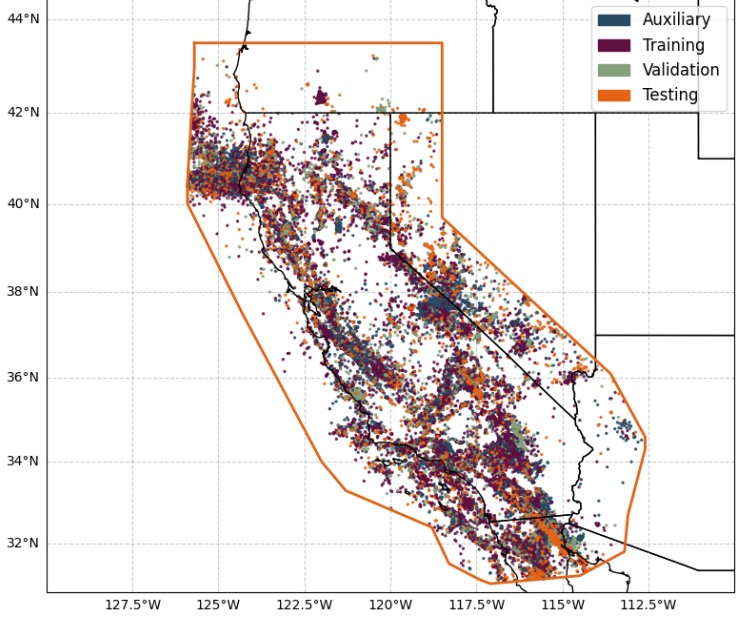

Figure 22: Locations of events in the `ComCat` dataset, labeled by their partition into auxiliary, training, validation and testing periods.

## H.2   SCEDC

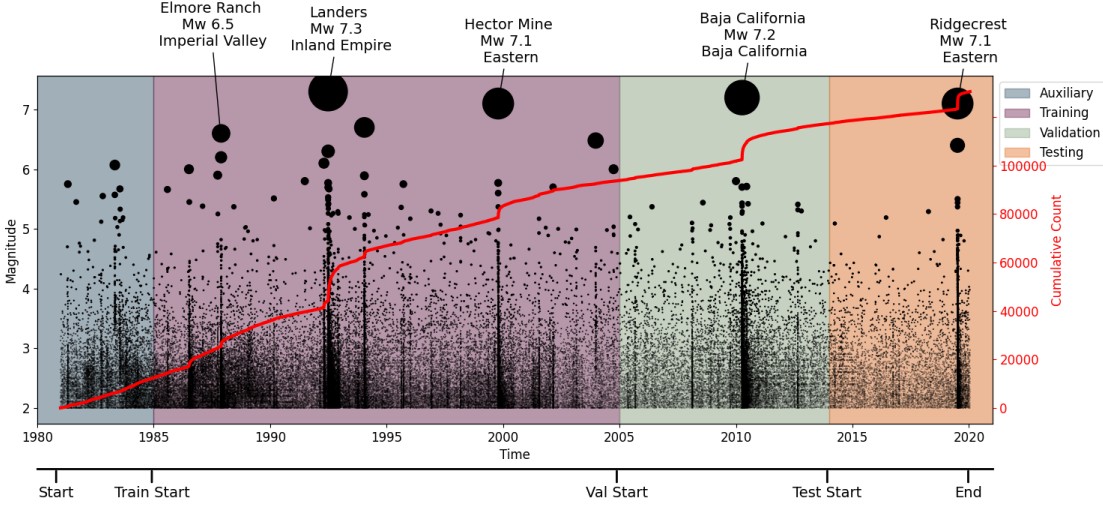

Figure 23: Times and magnitudes of events in the SCEDC dataset (with key events labeled). The size of the points are plotted on a log scale corresponding to Mw. Auxiliary, training, validation and testing periods are indicated by colour and a further cumulative count of events is indicated in red.

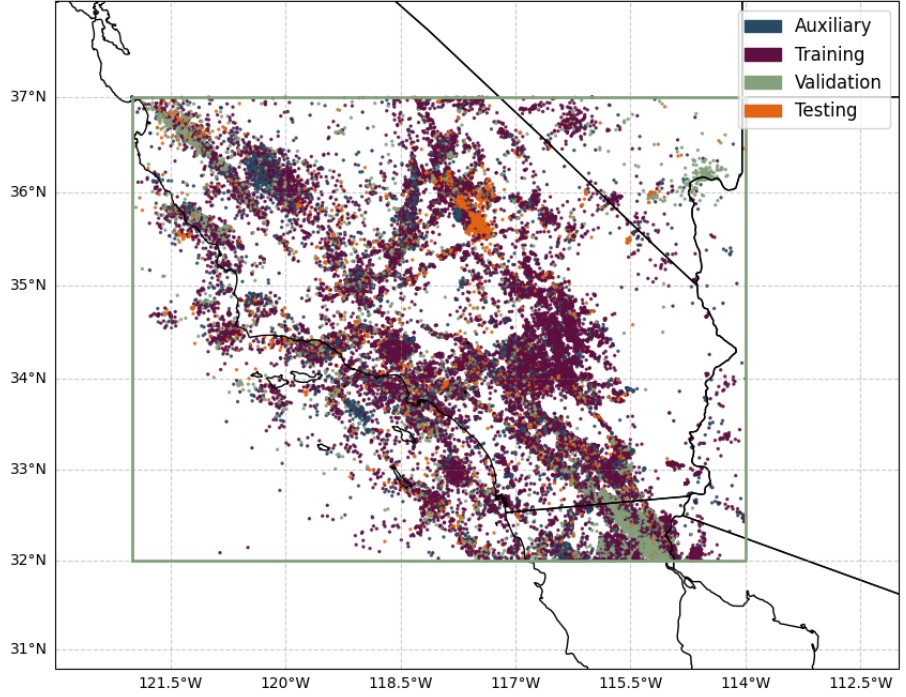

Figure 24: Locations of events in the SCEDC dataset, labeled by their partition into auxiliary, training, validation and testing periods.

### H.3 `White`

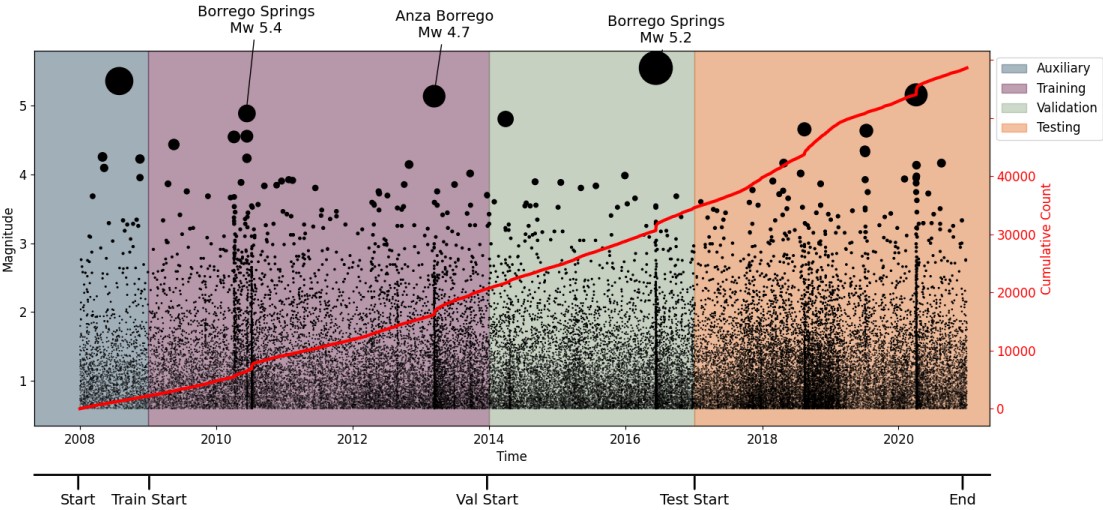

Figure 25: Times and magnitudes of events in the `White` dataset (with key events labeled). The size of the points are plotted on a log scale corresponding to Mw. Auxiliary, training, validation and testing periods are indicated by colour and a further cumulative count of events is indicated in red.

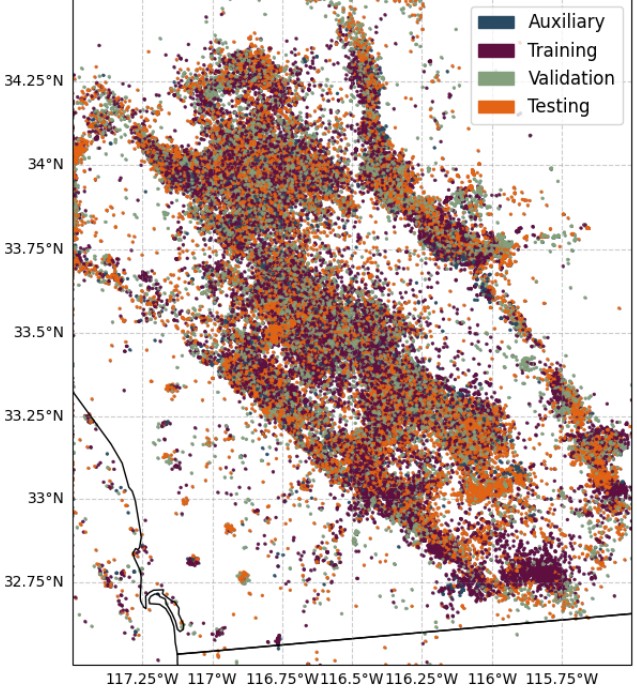

Figure 26: Locations of events in the `White` dataset, labeled by their partition into auxiliary, training, validation and testing periods.

### H.4 QTM_SanJac

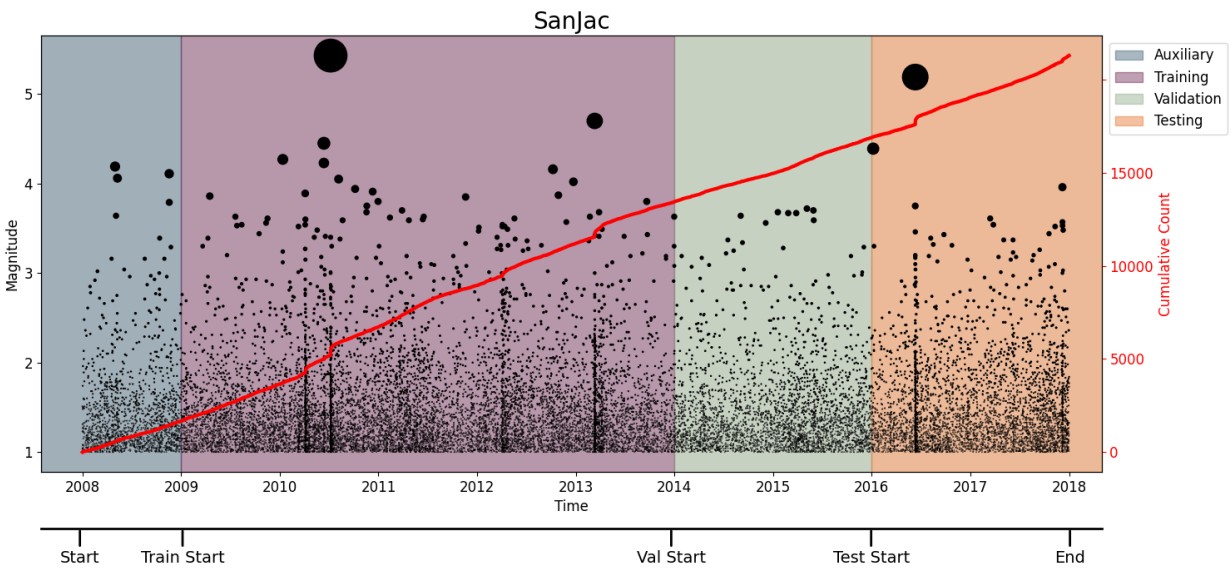

Figure 27: Times and magnitudes of events in the QTM_SanJac dataset. The size of the points are plotted on a log scale corresponding to Mw. Auxiliary, training, validation and testing periods are indicated by colour and a further cumulative count of events is indicated in red.

### H.5 QTM_SaltonSea

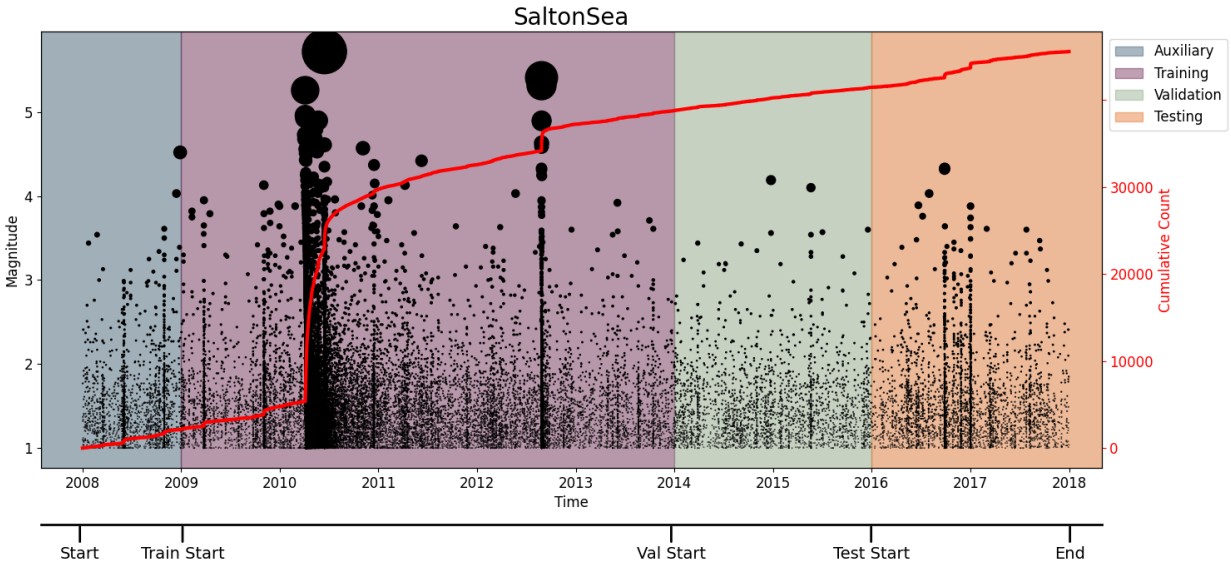

Figure 28: Times and magnitudes of events in the QTM_SaltonSea dataset. The size of the points are plotted on a log scale corresponding to Mw. Auxiliary, training, validation and testing periods are indicated by colour and a further cumulative count of events is indicated in red.

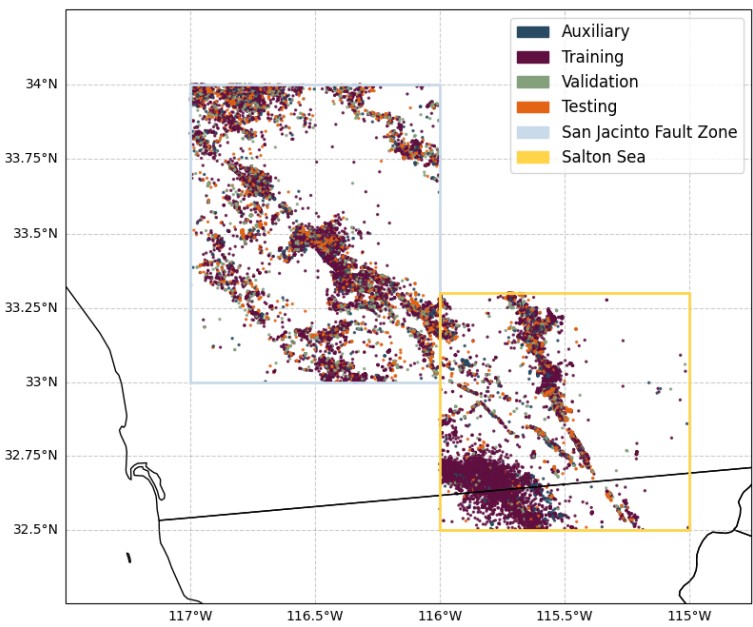

Figure 29: Locations of events in the `QTM_SanJac` and `QTM_SaltonSea` datasets, labeled by their partition into auxiliary, training, validation and testing periods.

## I  Error Distributions & Next-event metrics

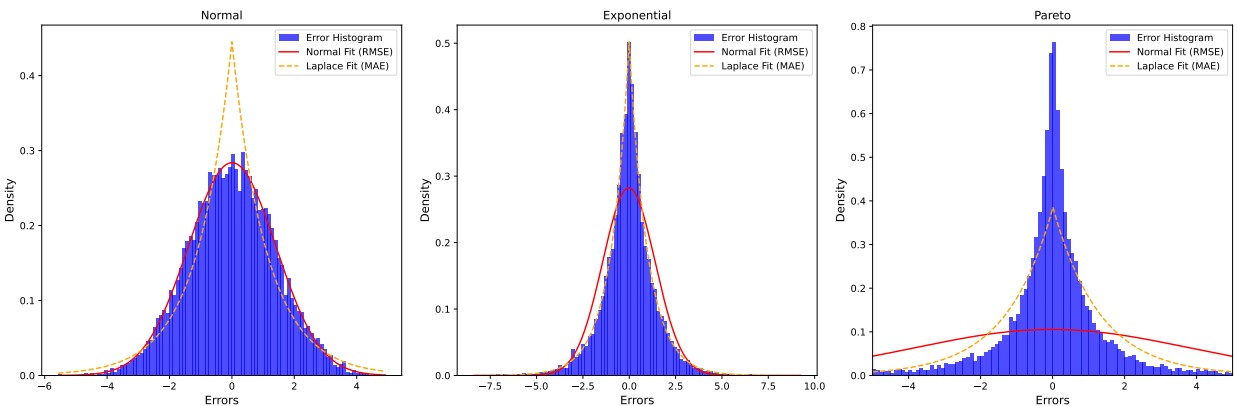

Figure 30: The distribution of errors ($Y_{\mathrm{obs}} - Y_{\mathrm{pred}}$) for the Normal$(0, 1)$, Exponential$(1)$, and Pareto$(2)$ distributions. Maximum likelihood estimation is used to fit Normal and Laplace distributions to each error histogram. Normal errors (Normal $\times$ Normal) are best approximated by the Root Mean Square Error (RMSE), while Laplacian errors (Exponential $\times$ Exponential) are best approximated by the Mean Absolute Error (MAE). However, neither RMSE nor MAE effectively capture the errors for the heavy-tailed Pareto distribution.

