# OpenReview forum: "EarthquakeNPP: A Benchmark for Earthquake Forecasting with Neural Point Processes"
_TMLR — Accepted by TMLR_

### Review · Reviewer_YgWk · 2025-11-11

**Summary Of Contributions:**

This paper introduces a large earthquake dataset designed for evaluating Neural Point Processes (NPPs). In contrast to previous datasets used in this area, the proposed datasets cover target regions ranging from small to large scales. The paper also compares recent NPPs with the classical ETAS model, demonstrating that ETAS performs comparably to, or even outperforms, the NPPs.

**Audience:**

Yes

**Audience Explanation:**

1.	Providing a comprehensive and large-scale dataset for evaluating Neural Point Processes is an important and timely contribution.
2.	The surprising result that the classical ETAS model performs comparably to—or even better than—modern Neural Point Processes provides new insights and may guide future research directions in this field.

**Claims And Evidence:**

Yes

**Claims Explanation:**

Although the discussion in this paper has room for improvement, I think its claims are supported by experimental evidence.

**Requested Changes:**

Major Points;

1.	The review of Neural Point Processes is severely lacking. The paper does not clearly explain the motivation behind introducing Neural Point Processes, making it difficult to fully appreciate the significance of the experimental results. I strongly encourage the authors to include a clear explanation of the motivation and development of Neural Point Processes, and to discuss why the previous datasets were insufficient to demonstrate this motivation.

2. The paper should elaborate on the insights and promising research directions inspired by the experimental findings. For instance, the authors say  in Section 5 that

“Designing NPPs with scalable, long-term memory mechanisms is therefore a critical avenue for improvement.”

However, it remains unclear how such mechanisms can be designed, or whether it is even feasible. Since this is one of the main implications of the paper, the authors are encouraged to elaborate on these points in more depth, with a clearer explanation of Neural Point Processes.


Minor Points:

3.. Abstract: “seismology.To address” → should be “seismology. To address”

4. Section 2.2: “(ta,tb) ∈R” → should be “(ta,tb)⊂R” ?

---

> ### Author Response · Authors · 2025-12-17
> **Response to Reviewer YgWk**
>
> We appreciate the reviewer’s recognition of the usefulness of the EarthquakeNPP dataset and the potential of our findings to inform future research. In response to all the requested changes, we have strengthened the manuscript by expanding the background on Neural Point Processes (NPPs) and by providing a clearer, more detailed discussion of promising future directions for NPP-based earthquake modelling.
>
>
> > "The review of Neural Point Processes is severely lacking."
>
> We agree that a clearer review of NPPs is essential for appreciating the motivation behind our benchmark and the significance of our findings. While the manuscript already discusses the limitations of the previous earthquake dataset in the section 1.1.1, this focuses on data issues (leakage, missingness) and benchmarking practices (lack of SOTA earthquake baseline) rather than the conceptual foundations of NPPs.
>
> To address the reviewer’s request, we have added a new subsection in the Background section (“2.3 Neural Point Processes”). This section provides a concise review of NPPs, including their motivation, modelling philosophy, main architectural classes, and their potential advantages and limitations for earthquake forecasting. This addition places our experimental results in clearer methodological context and strengthens the motivation for EarthquakeNPP.
>
> > "The paper should elaborate on the insights and promising research directions inspired by the experimental findings."
>
> We thank the reviewer for highlighting the importance of elaborating on the insights and future research directions arising from our experimental findings. While the original manuscript identified three key areas for improvement (magnitude modelling, scalable long-term memory, and generative evaluation mismatch), we agree that these were presented at a high level. In the revised version, we have substantially expanded Section 5 to make these directions more detailed, concrete, and actionable.
>
> 1. **Scalable long-term memory mechanisms**:
> Previously, we only stated that improved memory was needed. We now provide concrete strategies for how this could be implemented in practice. Specifically, we discuss the feasibility of long-context architectures inspired by Natural Language Processing (NLP), including sparse/dilated attention mechanisms for reducing O(N²) cost, hierarchical or coarse-to-fine encoding of earthquake histories, and event-history compression/summary memory modules to retain the influence of distant large events. We also clarify why this is feasible by referencing long-sequence transformer developments in the NLP literature and relate them directly to earthquake triggering dynamics.
>
> 2. **Magnitude-aware architecture design**:
> We expanded this direction to suggest modelling magnitude via hierarchical encoding of small vs. large events, magnitude-weighted attention, or parametrisations aligned with the log-frequency–magnitude scaling observed in seismicity. This provides a practical link between ETAS’s success and how NPPs may incorporate similar structure.
>
> 3. **Generative training aligned with CSEP evaluation**:
> We now clarify the training–evaluation mismatch for generative NPPs and propose solutions such as multi-event trajectory training objectives or losses targeting CSEP statistics directly, rather than single-step likelihood alone.
>
> 4. **Hybrid physics-informed kernels (new addition)**:
> We have also added a fourth future direction discussing how NPPs might incorporate physically motivated power-law kernels used in ETAS. While neural models offer flexibility, combining neural density estimation with ETAS-inspired scaling may yield architectures that capture aftershock clustering more faithfully than, e.g Gaussian kernel formulations used in DeepSTPP.
>
> To make these contributions clearer, we have formatted them as "Actionable Future Directions". We believe this expansion significantly strengthens the implications of our findings and addresses the reviewer’s concerns regarding clarity and feasibility.
>
> #### Typos
>
> > Abstract: “seismology.To address” → should be “seismology. To address”
>
> Thank you, this typo has been corrected.
>
> > Section 2.2: “(ta,tb) ∈R” → should be “(ta,tb)⊂R” ?
>
> Agreed, this interval is a subset not an element of \mathbb{R}. This has been corrected.

---

### Review · Reviewer_m3e9 · 2025-11-20

**Summary Of Contributions:**

Summary:

The authors provide EarthquakeNPP, a collection of public datasets together with a benchmark for evaluating neural point process (NPP) models on the task of earthquake forecasting.

They evaluate 5 different spatio-temporal NPPs, two of which are generative models, and compare their performance against ETAS (a model commonly used within the seismology community).

EarthquakeNPP consists of 4 different datasets, documenting earthquakes within California from 1971 to 2021.

They find that none of the 5 evaluated NPP models outperforms the ETAS baseline, indicating that _``current NPP architectures are not yet suitable for operational use''_. They also briefly discuss a few directions for how NPPs potentially could be improved based on their findings.




***
Strengths:
- The paper is quite well written and easy to follow overall.
- The studied problem is quite interesting, and I definitely think that EarthquakeNPP could be a useful resource for some parts of the seismology and machine learning communities.





***
Weaknesses:
- The technical contribution is limited. This is not necessarily an issue for TMLR, but still something that makes the paper less interesting than it could be. The paper would definitely be stronger if the authors also evaluated at least some type of potential model modification/improvement based on their findings and the discussion in Section 5.
- The model evaluation/comparison is not overly detailed, and the analysis could definitely be more in-depth. For example, the 5 NPP models are not compared in terms of computational cost, and the trade-off between performance and computational cost for the past event history could be analyzed.






***
Questions/suggestions:
- Section 3 is very short, you are not close to the 12-page limit, and the appendix contains quite a lot more information about the datasets etc. Would perhaps make sense to add a bit more details to the main paper?
- What is meant by SCEDC_20/25/30 in Figure 2 & 3? This is not explained (at least not in the main paper)?
- Could you perhaps comment on the performance relative to the Poisson baseline in Figure 2 & 3? For example, the Poisson performance is poor on SCEDC_30 but basically the same on White in Figure 2, does this make sense?
- In Figure 3, the performance of NSTPP on White seems to stand out. Does it make sense that this model performs poorly on this particular dataset? If so, why is its performance not poor on White also in Figure 2?
- You write that DeepSTPP and AutoSTPP condition on just the 20 past events. How is the performance affected if this conditioning window is decreased even further (e.g., to 10 or 5 past events)? And what happens if you increase it to 40? Or, is this not possible at all due to computational cost reasons?
- Section 5 is quite interesting, with the potential directions for how to improve NPPs. Could make sense to emphasize this more, e.g. by creating a list of "Main actionable takeaways" or similar?






***
Minor things:
- Abstract, "seismology.To address": typo.
- Abstract: "future collaboration between the seismology and machine learning" --> "future collaboration between the seismology and machine learning communities"?
- Section 1: "doesn’t compare against state-of-the-art models like ETAS" --> "does not compare against state-of-the-art models like ETAS"?
- 1.1.1: 'these sequences "too long" and "outliers." However, this' --> 'these sequences "too long" and "outliers". However, this'.
- 1.1.2: "models that don’t output" --> "models that do not output"?
- Table 1: "casts they release to the public.)" --> "casts they release to the public).".

**Additional Comments:**

No additional comments.

**Audience:**

Yes

**Audience Explanation:**

The studied problem is quite interesting, and I think EarthquakeNPP could be a useful resource.

**Broader Impact Concerns:**

No concerns.

**Claims And Evidence:**

No

**Claims Explanation:**

The current model evaluation/comparison is not quite detailed enough, the analysis is a bit too shallow. See "Weaknesses" and "Questions/suggestions" above.

**Requested Changes:**

This is a quite well-written paper overall that I think could be relevant for the TMLR audience. I definitely think that EarthquakeNPP could be a useful resource.

However, I think the current version requires some clarifications and tweaks, see "Weaknesses" and "Questions/suggestions" above.

---

> ### Author Response · Authors · 2025-12-17
> **Response (1/2) to Reviewer m3e9**
>
> We thank the reviewer for their constructive and thorough review, particularly for prompting us to greatly expand the detail of our model comparison - this has significantly improved the paper. We have responded to all your points below, omitting the minor corrections for lack of space.
>
> > The model evaluation/comparison is not overly detailed, and the analysis could definitely be more in-depth. For example, the 5 NPP models are not compared in terms of computational cost...
>
> Thank you for this constructive comment. We agree that the original model evaluation could be more in-depth, particularly in terms of understanding *why* models succeed or fail. While the initial submission included additional analysis figures in the appendix (original Figures 11, 12, and 14), we agree that their implications were not sufficiently discussed.
>
> We note that a comparison of computational cost was already provided in Appendix D in the original manuscript. Following this comment and the related suggestions of Reviewer RSG5, we have substantially expanded this analysis to include inference (simulation) cost in addition to training cost, and we now explicitly reference both from Section 4. This extension is particularly relevant for operational forecasting, where simulation efficiency is critical.
>
> To provide a more detailed and informative model comparison, we have made the following major additions:
>
> 1. **Expanded likelihood analysis (Appendix E).**
>    Appendix E, which previously presented only time-series plots of information gain relative to ETAS, has been expanded to include a detailed discussion of model performance across time and across all datasets. We additionally include new figures showing the spatial distribution of log-likelihood scores, which reveal systematic differences between NPP architectures, including degraded performance during large mainshock–aftershock sequences and improved performance in complex tectonic areas and away from faults.
>
> 2. **New CSEP analysis section (Appendix G).**
>    We introduce a new appendix section providing a detailed analysis of the CSEP consistency tests. This section includes daily temporal diagnostics and aggregated spatial forecasts for ETAS, SMASH, and DSTPP, enabling direct comparison of underprediction, overprediction, and spatial bias across models. These results clarify the mechanisms behind the pass rates reported in the main text and demonstrate how temporal variability and spatial smoothness differ across generative approaches.
>
> Together, these additions substantially deepen the evaluation by connecting quantitative metrics to model behaviour in time, space, and computational efficiency.
>
> > The paper would definitely be stronger if the authors also evaluated at least some type of potential model modification based on their findings... How is the performance affected if this conditioning window is decreased or increased?
>
> We appreciate the reviewer’s suggestion that evaluating any of our suggested improvements would strengthen the contribution of this paper. We agree that exploring our suggested architectural changes (e.g., longer memory, magnitude inclusion, power-law kernels) is an important and natural next step, and we have now expanded Section 5 to outline actionable research directions arising from our experimental findings.
>
> However, we respectfully argue that implementing and evaluating such model developments falls beyond the intended scope of this paper. The primary goal of this work is:
>
> _to direct NPP modeling away from the existing flawed benchmark dataset (Chen et al., 2021) by presenting an appropriate alternative dataset (EarthquakeNPP), and demonstrating that a seismology model (ETAS) currently surpasses any existing implementations of NPPs on earthquakes._
>
> To ensure a fair comparison and isolate the effect of the benchmark itself, all NPP hyperparameters were kept consistent with their existing implementations on the Chen et al. (2021) earthquake dataset. While (as you suggest) increasing the conditioning window is computationally costly ($O(n^2)$), our primary reason for not pursuing such variations is that they constitute model exploration rather than benchmark evaluation. We hope that EarthquakeNPP provides the platform for the essential next step: **exploring how to improve the performance NPP models for the task of earthquake forecasting.**

---

> ### Author Response · Authors · 2025-12-17
> **Response (2/2) to Reviewer m3e9**
>
> > Would perhaps make sense to add a bit more details to the main paper?"
>
> We agree that bringing content from the appendix into the main text strengthens Section 3. Following your suggestion, we have now included a discussion on the data missingness issue in Section 3.
>
> > What is meant by SCEDC_20/25/30 in Figure 2 & 3?
>
> Apologies, we tried to make this explicit in the SCEDC column of table 1. "SCEDC_20/25/30" refers to 3 magnitude thresholds of the raw SCEDC catalog at magnitudes (2.0,2.5,3.0). We have updated table 1, Figure 2 and Figure 3 with a reminder of this.
>
> > Could you perhaps comment on the performance relative to the Poisson baseline in Figure 2 & 3? For example, the Poisson performance is poor on SCEDC_30 but basically the same on White in Figure 2, does this make sense?
>
> The relative Poisson performance in Figures 2 reflects the level of temporal clustering in each dataset. When seismicity is weakly clustered and closer to a homogeneous Poisson process, the Poisson baseline performs similarly to the other models (e.g., WHITE, QTM_SanJac). In contrast, datasets with strong temporal clustering (e.g., QTM_SaltonSea) deviate more from Poisson, leading to poorer Poisson likelihood scores. The amount of clustering is visible in the cumulative event curves in Appendix H, where clustered catalogs show sudden increases in event counts. An equivalent comparison can be made with figure 3 and the amount of spatial clustering present in each dataset.
>
> This discussion has been incorporated into section 4.1.
>
> > In Figure 3, the performance of NSTPP on White seems to stand out. Does it make sense that this model performs poorly on this particular dataset? If so, why is its performance not poor on White also in Figure 2?
>
> Thank you for your question, we were also confused by the poor performance of NSTPP on the White dataset. The honest answer is that we don't know exactly why the performance is so poor, however your question has prompted us to hypothesise training instability as the possible cause:
>
> Since the White dataset contains the lowest magnitude data this means that the locations of earthquakes contained in this dataset are closest together. This may have an affect on NSTPP training if the model perceives two events to be at the same location (within some tolerance) and could result in some type of division/multiplication by $\epsilon$. Since the temporal and spatial components of NSTPP are conditionally independent, $ \lambda(t,x) = \lambda(t)p(x|t) $, poor spatial performance does not imply poor temporal performance.
>
> We have included our hypothesis as to the poor spatial performance in in the new log-likelihood analysis section E.2.
>
> > Section 5 is quite interesting, with the potential directions for how to improve NPPs. Could make sense to emphasise this more, e.g. by creating a list of "Main actionable takeaways" or similar?
>
> We are glad you enjoyed this section. Your comment and the suggestion of reviewer YgWk have prompted an expansion of future directions discussion in Section 5. Notably we have:
>
> 1. Expanded on the our original three suggested improvements with more concrete suggestions of model changes.
> 2. Added a fourth suggestion relating to power-law behaviour regularly observed in seismic data.
> 3. Formatted these as a list of "Actionable Future directions".

---

> > ### Comment · Reviewer_m3e9 · 2026-01-05
> >
> > Thank you for the response.
> >
> > I have read the other reviews and all author responses.
> >
> > The other reviews are quite positive overall. The authors provided a solid rebuttal that mostly addresses my concerns.
> >
> > This is a quite well written and solid paper overall, and I definitely think that EarthquakeNPP could be a useful resource. I will therefore recommend accept.

---

### Review · Reviewer_RSG5 · 2025-12-04

**Summary Of Contributions:**

This paper introduces EarthquakeNPP, a benchmark to evaluation neural point processes on seismic data.
This benchmark includes multiple datasets on California earthquakes publicly available, as well as domain specific baselines (ETAS) and evaluation protocols tailored to the seismic domain.
The main evaluation is based on the classical log-likelihood on a separate test set, as well as seismic specific evaluation CSEP, which use 10000 simulations and compute various statistical tests.
The key conclusion is that none of the existing neural point process models outperform the ETAS model on this benchmark, demonstrating the need for further research in this area, and offering a reliable way to assess the progress of future models.

**Additional Comments:**

- Abstract: `seismology.To` -> `seismology. To`
- *"relied upon refinement of [...] ETAS model, despite significant growth in available data"*:: It is unclear why the growth in available data would imply that ETAS should be outperformed by other models. This could be clarified, saying that this is a simple model, which may be limited in its expressiveness compared to neural networks.
- NPP references: Neural Hawkes Process [A] would be a good reference to include in the intro.
- Recent papers proposed parametric models for seismic data that could be mentioned in the related work, e.g., [B], [C].
- Figure 2 and 3: It would be helpful to reorder the methods, to have the three NPP models grouped together, and ETAS/Poisson baselines together.

**References:**

[A] Mei, H., and Eisner, J. ["The neural hawkes process: A neurally self-modulating multivariate point process."](https://arxiv.org/abs/1612.09328) NeurIPS (2017).
[B] Bernabeu A, Mateu J. ["Spatio-temporal Hawkes point processes: statistical inference and simulation strategies"](https://arxiv.org/pdf/2511.14509). arXiv (2025).
[C] Siviero, E., Staerman, G., Clémençon, S. & Moreau, T. ["Numerically Efficient Parametric Inference for Learning Space-Time Hawkes Processes"](https://openreview.net/pdf?id=VqfvOXyTPj). DSAA (2025).

**Audience:**

Yes

**Audience Explanation:**

It is somewhat relevant to TMLR audience that is developing spatio temporal PP models but may be missing some more complete evaluation, and some code to easily extend this benchmark.

**Claims And Evidence:**

Yes

**Claims Explanation:**

- **S1:** The exposition of the paper is clear and easy to follow.
- **S2:** The purpose of the benchmark is well motivated, and I think this is a step in the right direction to show that usual benchmarks can be misleading for real world applications. The performances of classical tools like ETAS need to be included in benchmarks to give the full picture on the performance of the model.
- **S3:** The evaluation protocol, based on two metrics is well justified. The addition of CSEP is relevant for the seismic domain.

---

- **W1:** One weakness of the benchmark is the lack of common evaluation for all methods. In particular, likelihood is only reported for some methods, and CSEP only for others. This makes to completely grasp the strength and weaknesses of the method (See **C1** and **C2**).
- **W2:**  Moreover, the evaluation proposed in the code is not centralized, making it very error prone (**C3**). The code for the evaluated methods is also not present in the supplementary materials. Overall, I feel more care should be spend on designing the benchmark to make it easy to use and extend for future methods, with common objects to load data and common evaluation protocol.

**Requested Changes:**

- **C1:** While I understand that simulation might be slow with DeepSTPP, AutoSTPP, and NSTPP, I think it should be run at least on the smallest dataset, to show which order of magnitude this is. As these models relies on Hawkes model, it seems it is possible to run immigration birth algorithm for simulation, which should be faster than thinning. But even if it is not possible, evaluating on a small dataset would be useful to have a more complete comparison of the methods. For the SMASH method, the method seems able to predict magnitude but has been used with quantized magnitude in the original paper (small/medium/large). Wouldn't it be possible to have predict magnitude directly with the method? For the NLL metric, two methods are excluded. However, looking at the papers introducing these methods, DSTPP uses a variational lower-bound to estimate the NLL. This could also provide some insights on the performance of the DSTPP and SMASH models.
Overall, I think having at least one common metric (possibly on a subset of the datasets for the slowest methods, or using approximations such as variational bounds) would be useful to make this benchmark more complete.
- **C2:** I appreciate the fact that the authors report training time for the models. Adding runtime evaluation for inference would also be a useful addition to the benchmark, as in real world applications, inference time is often a bottleneck.
- **C3:** The code provided does not include the methods other than ETAS. From the `README.md`, it seems that each method uses it own script to run. This should be improved as I think the benchmark would benefit from a more modular codebase, with common data loaders and evaluation protocols for all methods. With the current design, it is very error prone to add a new method, as a complete script needs to be written. Having a more modular approach would make it easier to extend the benchmark in the future, and avoid errors in the evaluation. I would recommend to have a common evaluation script that would take as input the predictions of the models, and compute all the metrics (NLL and CSEP). Note that some python packages such as [`benchopt`](https://benchopt.github.io/) provide dedicated tools to make it easier to build reusable benchmarks.

---

> ### Author Response · Authors · 2025-12-17
> **Response (1/2) to Reviewer RSG5**
>
> We thank the reviewer for highlighting the importance of the benchmark’s motivation, especially the need to include established models such as ETAS to avoid misleading conclusions about neural model performance in real-world applications. We address all your comments below, omitting our minor corrections due to lack of space.
>
> > One weakness of the benchmark is the lack of common evaluation for all methods. In particular, likelihood is only reported for some methods, and CSEP only for others.
>
> We appreciate this point and agree that evaluating all five NPPs and ETAS under a fully unified set of metrics would make the benchmark even more complete. However, as described below, differences in model formulation and simulation feasibility prevented us from applying both likelihood and CSEP metrics to all models without re-engineering them. Importantly, despite the partial metric coverage, **all available comparisons consistently show ETAS outperforming the NPPs**, which we see as the key, unifying result and a performance target for future earthquake-focused NPP research.
>
> > While I understand that simulation might be slow with DeepSTPP, AutoSTPP, and NSTPP, I think it should be run at least on the smallest dataset... As these models relies on Hawkes model, it seems it is possible to run immigration birth algorithm for simulation, which should be faster than thinning. But even if it is not possible, evaluating on a small dataset would be useful to have a more complete comparison of the methods."
>
> To begin with, **NSTPP is not Hawkes-based**, and generating events requires solving an ODE for each arrival, making simulation extremely slow even for small windows. For **DeepSTPP and AutoSTPP**, although they draw inspiration from Hawkes processes, **their non-stationary neural triggering kernels depend on the full event history**, meaning immigration–birth sampling cannot be applied in the standard way used for ETAS. Unlike classical Hawkes models, new events cannot be generated in parallel because the triggering kernel changes after every event. To our knowledge, **no fast immigration–birth sampling method exists for these architectures.**
>
> Furthermore, thinning is also not straightforward. AutoSTPP does not enforce monotonic or decaying kernels, so $\lambda^*(t,x)$ cannot be safely upper-bounded, a requirement for Ogata thinning, making robust simulation non-trivial. While DeepSTPP _can_ be simulated by thinning, the approach would be incredibly slow. Our estimates for simulating from the model on the smallest dataset SanJac, would require on the order of 350,000 CPU hours. We have included these limitations about AutoSTPP and DeepSTPP in the Appendix section D.2 "Simulation".
>
> > "For the SMASH method, the method seems able to predict magnitude but has been used with quantized magnitude in the original paper (small/medium/large). Wouldn't it be possible to have predict magnitude directly with the method?"
>
> We thank the reviewer for this suggestion. We did initially attempt to extend SMASH to predict magnitudes in finer bins aligned with the CSEP evaluation (0.1 magnitude units rather than the original small/medium/large discretisation). However, in our experiments this modification led to a deterioration in spatio-temporal performance metrics during training, negatively affecting both time and spatial predictions. Fully diagnosing or redesigning the magnitude component would require additional model development beyond the scope of this benchmark study. For the present work, we therefore restrict SMASH to rate forecasting only, and we now clarify this design choice in Appendix Section F.4 of the manuscript.

---

> ### Author Response · Authors · 2025-12-17
> **Response (2/2) to Reviewer RSG5**
>
> > For the NLL metric, two methods are excluded. However, DSTPP uses a variational lower-bound to estimate the NLL.
>
> This is a great suggestion, and one that we tested previously. The table reports the DSTPP variational lower-bound likelihoods (mean ± std) alongside AutoSTPP, a likelihood-based NPP evaluated with the same metric. Across all datasets, DSTPP scores are substantially lower than AutoSTPP, indicating that **the variational bound is not tight in this setting**. For this reason, we omit DSTPP from the main likelihood comparison in the manuscript, as the resulting values are not directly comparable to exact likelihood estimates.
>
>
> | Dataset        | Model     | Temporal LL  | Spatial LL  |
> |----------------|-----------|------------------:|-----------------:|
> | ComCat         | AutoSTPP  | **1.496 ± 0.925**     | **−9.852 ± 0.066**   |
> | ComCat         | DSTPP     | 1.135 ± 0.256     | −16.220 ± 0.027  |
> | SaltonSea  | AutoSTPP  | **2.006 ± 0.481**     | **−2.938 ± 0.003**   |
> | SaltonSea  | DSTPP     | 0.768 ± 0.377     | −10.632 ± 0.189  |
> | SanJac     | AutoSTPP  | **1.070 ± 0.000**     | **−6.257 ± 0.000**   |
> | SanJac     | DSTPP     | 0.180 ± 0.175     | −15.801 ± 0.319  |
> | SCEDC_20       | AutoSTPP  | **2.849 ± 0.927**     | **−8.428 ± 0.018**   |
> | SCEDC_20       | DSTPP     | 0.093 ± 0.086     | −12.960 ± 0.060  |
> | White          | AutoSTPP  | **1.969 ± 0.005**     | **−5.282 ± 0.011**   |
> | White          | DSTPP     | 1.461 ± 0.314     | −16.648 ± 0.676  |
>
>
> > Adding runtime evaluation for inference would be a useful addition to the benchmark
>
> Thank you for this suggestion, we agree that inference runtimes would be useful and have updated appendix D "Computational Efficiency" to include them.
>
> > evaluation proposed in the code is not centralised.....I think the benchmark would benefit from a more modular codebase, with common data loaders and evaluation protocols for all methods. With the current design, it is very error prone to add a new method, as a complete script needs to be written.
>
> We thank the reviewer for this valuable suggestion. We agree that a modular codebase with unified data loading and evaluation would enhance usability and make future extensions less error-prone. In the present work, however, we made a deliberate choice _not_ to re-implement the NPP models within a common training/evaluation framework, as this would require substantial engineering effort and risk introducing _errors_ or inconsistencies relative to the original papers. Instead, we prioritised evaluating existing implementations faithfully (using the authors own scripts), to reflect the current state of NPP performance on earthquakes without any modifications.
>
> Our primary objective in this paper is to address a more fundamental issue: **current practice in the NPP community relies heavily on the Chen et al. (2021) benchmark, which contains data leakage and omits key earthquakes**, leading to potentially misleading conclusions about model performance. EarthquakeNPP is intended as a corrective step, providing a clean, seismologically grounded dataset and demonstrating that ETAS remains a strong baseline under realistic evaluation. We see this as a necessary foundation before a unified modelling library can be sensibly built.
>
> **With respect to evaluation design**, we would like to clarify that **CSEP evaluation is already centralised** in the repository and can be run through a single script ([link](https://anonymous.4open.science/r/EarthquakeNPP--anon--C444/Experiments/run_pycsep_tests.py)) once forecasts are generated. The challenge arises specifically for **log-likelihood evaluation**, where a common interface is not possible due to model architecture differences;
>
> This is because **likelihood is defined by the parameterisation of each model:**
>
> * **DeepSTPP and AutoSTPP** pose a conditional intensity $\lambda^*(t,x)$, allowing likelihood computation directly;
> * **NSTPP** does not define an explicit intensity, instead defining its own likelihood through continuous-time normalising flows;
>
> Creating a unified likelihood evaluation pipeline would therefore require substantial model re-implementation rather than a wrapper-level change. Instead, to support extensibility, we have added **a worked example notebook showing how to add a new model** [here](https://anonymous.4open.science/r/EarthquakeNPP--anon--C444/Experiments/guide_for_new_models.ipynb), including where likelihood evaluation should be inserted if the model provides intensity outputs. We hope this addition helps researchers integrate further architectures into the benchmark more easily.
>
> > The code provided does not include the methods other than ETAS.
>
> Apologies, this was an unforeseen issue when annonomising a GitHub repository that uses "submodules". We have now updated the supplied code to include all other models [(link)](https://anonymous.4open.science/r/EarthquakeNPP--anon--C444/README.md).

---

### Author Response · Authors · 2025-12-17
**Revised Manuscript Uploaded by Authors**

We would like to thank the reviewers for their time and constructive feedback.

We apologise for the delay in submitting our responses; during this time we have made substantial improvements to the manuscript, including:

- an **expanded likelihood analysis (Appendix E)**, adding detailed discussion of model performance across time and datasets and **new spatial log-likelihood figures** that reveal systematic differences between model outputs;

- a **new CSEP analysis section (Appendix G)**, providing detailed **temporal and spatial diagnostic plots** for ETAS, SMASH, and DSTPP that clarify the sources of underprediction, overprediction, and spatial smoothness;

- a **new review section 2.3** on NPPs;

- an extended discussion section providing **actionable future directions**;

- a new section describing the **computational cost of inference**; and

- a **worked example notebook** demonstrating how to add a new model, available [here](https://anonymous.4open.science/r/EarthquakeNPP--anon--C444/Experiments/guide_for_new_models.ipynb).

In particular, examining the temporal output of the simulated forecasts allowed us to identify a rounding error in our implementation (using “≤” instead of “<”). Correcting this issue leads to marginally improved pass rates for the CSEP number test across all models, without altering any model rankings or the conclusions of the study.

---

### Public Comment · ~Ranjit_Das1 · 2026-07-19
**Need for Validation of Earthquake Catalog Preparation and Magnitude-Scale Consistency in Seismic Forecasting**

The manuscript does not provide sufficient details regarding the preparation, and validation of the earthquake catalog used for the forecasting analysis. It has been observed that Mw scale has been used for smaller medium and large earthquakes.  A reliable and physically consistent earthquake catalog is a fundamental requirement for any seismic forecasting study. In this context, the exclusive use of the moment magnitude scale (Mw) requires further justification, particularly for magnitudes below approximately 7.5.
The M scale of Hanks and Kanamori (1979), which used Equation (1) of Purcaru and Berckhemer (1978), was originally derived based on the relationship between surface-wave magnitude (Ms) and seismic moment (M0), with applicability primarily limited to Ms≲7.0.  Hanks and Kanamori (1979) referred the equation as 5.0≲Ms≲7.5 in the derivation of M scale. Thus, use of M scale below 7.5 needs further clarifications (See Das et al, 2025). Recent studies, including Matsura (2025)., have also highlighted the necessity of revisiting conventional magnitude scale M and applying corrections to improve magnitude consistency.
I recommend that the authors investigate the impact of using an energy-consistent magnitude scale, such as the Das magnitude scale (Mwg; Das et al., 2019, BSSA; Das et al., 2026), for their catalog and forecasting analysis. Incorporating Mwg may significantly the forecasting results and conclusions of the manuscript.
A fundamental requirement of any magnitude scale is the consistency and compatibility among different magnitude measures (e.g., mb, Ms, and Mw), as emphasized by Kanamori (1977) and Hanks and Kanamori (1979) and Das et al (2019). However, recent investigations have demonstrated significant deviations between Mw and other magnitude scales, particularly when comparing Mw with mb and Ms (see Figures 1–2 of Das et al., 2019 and Figures 1–2 of Das et al., 2025; Das and Das , 2026). Furthermore, comparisons between radiated seismic energy and Mw indicate substantial inconsistencies at the global scale (see Figure 4 of Das et al., 2019, Figure 2 of Das et al ,2025).
Therefore, the authors should provide clear scientific justification for why the conventional Mw scale is appropriate below 7.5 for the present analysis. Specifically, they should discuss whether the limitations of Mw at smaller and moderate magnitudes could introduce bias in the earthquake catalog and affect the forecasting performance. Without addressing this issue, the reliability of the reported results and conclusions remains uncertain.

References
	Hanks TC, Kanamori H (1979) A moment magnitude scale. J Geophys Res 84:2348–2350
	Kanamori H (1977) The energy release in great earthquakes. J Geophys Res 82:2981–2987
	Purcaru G, Berckhemer H (1978) A magnitude scale for very large Earthquakes. Tectonophysics 49:189–198 12. Keir D, Stuart GW, Jackson A, Ayele A (2006)
	Das R, Sharma ML, Wason HR, Choudhury D, Gonzales G (2019) A seismic moment magnitude scale. Bull Seism Soc Am 109(4):1542–1555. https:// doi. org/ 10. 1785/ 01201 80338 9. Das R, Menesis C, Urrutia D (2023) Regression relationships
	Das, R., Das, A., Choudhury, D., Meneses, C., Barrera, F. G., Wason, H. R., & Alam, M. A. (2025). Updated probabilistic seismic hazard assessment using Mwg (Das magnitude scale) to address moment magnitude (M) scale inaccuracies below 7.5: a case study from Northeast India. Scientific Reports.
	Das, R., & Das, A. (2026). Limitations of Mw and M scales: Compelling evidence advocating for the das magnitude scale (Mwg)—A critical review and analysis. Indian Geotechnical Journal, 56(2), 979-997.
Matsu’ura, M. (2025). A theoretical basis of the moment magnitude scale. Earth, Planets and Space, 77(1), 151.

---

### Decision · Action_Editor_EUwH · 2026-02-01

**Recommendation:** Accept as is

**Additional Comments:**

Although the paper is acceptable in its current form, I strongly encourage the authors to take into account the recommendations provided by one of the reviewers. In particular, the comments in the review highlight the need for a sustainable benchmarking platform to have a test suite, a maintenance plan, and thorough, coherent documentation.

**Audience:**

Yes

**Audience Explanation:**

The benchmarking platform has the potential to be a useful resource for machine learning researchers exploring point processes and potential seismology applications.

**Claims And Evidence:**

Yes

**Claims Explanation:**

The paper makes the following claims:
(1)	It introduces a benchmarking platform that curates and standardizes multiple existing public resources;
(2)	The platform facilitates meaningful forecasting experiments relevant to stakeholders in the seismology community
(3)	The platform includes an operational-level implementation of the ETAS model and incorporates the generative evaluation procedures used in seismology.

One reviewer raised residual concerns about some of these claims, specifically emphasizing that the software component of the submission does not meet the expected standard of a benchmarking platform. While my overall assessment, taking into account the other reviews, is that the paper does provide suffcient evidence for its claims, I strongly recommend that the authors endeavour to make adjustments to their code to address the reviewer’s well-explained misgivings.